# Hunting for Gravitational Quantum Spikes

Andrzej Góźdź [1,*,†] , Włodzimierz Piechocki [2,†] , Grzegorz Plewa [2,†] and Tomasz Trześniewski [3,†]

[1] Institute of Physics, Maria Curie-Skłodowska University, pl. Marii Curie-Skłodowskiej 1, 20-031 Lublin, Poland

[2] Department of Fundamental Research, National Centre for Nuclear Research, Pasteura 7, 02-093 Warsaw, Poland; wlodzimierz.piechocki@ncbj.gov.pl (W.P.); grzegorz.plewa@ncbj.gov.pl (G.P.)

[3] Institute of Theoretical Physics, Jagiellonian University, Łojasiewicza 11, 30-348 Kraków, Poland; t.trzesniewski@uj.edu.pl

\* Correspondence: andrzej.gozdz@umcs.lublin.pl

† These authors contributed equally to this work.

**Abstract:** We present the result of our examination of quantum structures called quantum spikes. The classical spikes that are known in gravitational systems, occur in the evolution of the inhomogeneous spacetimes. A different kind of spikes, which we name strange spikes, can be seen in the dynamics of the homogeneous sector of the Belinski–Khalatnikov–Lifshitz scenario. They can be made visible if the so-called inhomogeneous initial data are used. The question to be explored is whether the strange spikes may survive quantization. The answer is in the affirmative. However, this is rather a subtle effect that needs further examination using sophisticated analytical and numerical tools. The spikes seem to be of fundamental importance, both at classical and quantum levels, as they may serve as seeds of real structures in the universe.

**Keywords:** classical gravity; quantum gravity; source of spacetime inhomogenity





## 1. Introduction

The dynamics underlying the Belinskii–Khalatnikov–Lifshitz (BKL) scenario, which concerns a generic gravitational singularity [1,2], can be described by the nonlinear coupled system of ODEs for the three effective directional scale factors (see Part I of [3]). These dynamics have been recently quantized [4,5]. The quantum BKL scenario predicts that a gravitational singularity can be avoided by a quantum bounce, occurring in the unitary evolution of a given gravitational system.

A different approach to solve the problem of a singularity in the BKL scenario has been proposed by Ashtekar et al. [6]. It can be used, after the successful quantization, to tackle a generic gravitational singularity as well. Furthermore, even if one restricts to its homogeneous sector, the model can be explored from the perspective of another interesting issue, which is the emergence of gravitational structures known as spikes. The aim of this paper is to uncover such structures at the classical level and to investigate if they can survive the quantization. The spikes that we name here the "strange spikes" are different from the (transient or permanent) spikes observed in the dynamics of inhomogeneous spacetimes (see [7–15] and references therein). The latter have well-understood properties, whereas our spikes have been discovered and preliminarily examined in the context of quantum physics only recently, in [16]. Results of the latter paper suggest that quantum (strange) spikes do not exist. However, the issue of time has not been treated satisfactorily due to the fact that [16] deals mainly with the vacuum case. In the present paper, we couple the system to a massless scalar field, so that it can be used as a reference clock at both the classical and quantum levels. Moreover, [16] has included only simplified analyses of quantum observables of a spike. Our paper fills this gap as well.

Let us stress that we do not address the issue of possible resolution of a generic gravitational singularity, which is predicted by the BKL conjecture and follows from the

quantization of the full classical dynamics. Instead, we examine the possibility of the formation of spikes resulting from the nonlinearity of the dynamics that is specific to its homogeneous sector, as defined in [6] by Equations (5.7)–(5.11). The latter is the total dynamics that are intended to be the subject of our paper.

The classical and quantum spikes that we examine are subtle structures, which appear in rather complicated dynamics at both the classical and quantum levels. They are of fundamental importance as they may serve as seeds of macro-structures in the Universe (like, e.g., filaments built from superclusters of galaxies) in the former case, and quantum fluctuations (which may underly, e.g., the creation of primordial black holes) in the latter case.

On the other hand, such structures are specific to any nonlinear coupled system of ODEs in which one considers the mapping of a smooth curve of the initial data into another curve via the propagation by the same amount of time at each point of the initial data. It may happen that the initial curve evolves into an the intriguing structure that we call a strange spike.

Spikes occur in a variety of dynamical systems. For instance, in the context of dynamics of a forced pendulum with damping. The damped driven (forced) pendulum models have applications in mathematical biology (see [17,18] and references therein). In general, the name "spike" is used in literature in very different physical and mathematical contexts. Actually, in most cases when the function describing a given phenomenon has a jump in some region, the latter is called a spike. An interesting problem is the existence of quantum spikes. The paper by A. Tilloy et al. [19] contains a particular example of them. The authors define quantum spikes as a certain kind of quantum fluctuations in the system that are able to jump between different states and can be described by a set of stochastic equations. This type of spikes, which probably can be observed in any quantum system satisfying required conditions, is different from quantum gravitational spikes presented in our paper.

As we already mentioned, an important aspect of this work is the problem of time, which occurs when quantization is applied to variables describing the dynamical spacetime geometry. At the classical level, we use the well-known formalism of parameterizing time via a scalar field, which acts as a reference clock. We propose to choose the corresponding reference clock at the quantum level, which requires introducing some special mathematical structure. This construction leads to a specific formalism describing the quantum dynamics. All details are presented in Section 3.

The paper is organized as follows: In Section 2, we define the phase space variables satisfying the affine Lie algebra, Hamilton dynamics parameterized by a massless scalar field, and derive the classical spikes. Section 3 deals with the quantization. We specify the representation of the Lie algebra and introduce the quantum evolution parameter. We represent the quantum evolution in terms of two eigenequations and find numerical solutions to these equations. The quantum dynamical constraint is imposed. In Section 4 we derive the quantum spikes. Conclusions are presented in Section 5.

## 2. Classical Level

### 2.1. Phase Space

The kinematical phase space of the homogeneous sector of the gravitational field in the Hamiltonian formalism [6] can be parameterized by the variables $C_I$ and $P_I$, with $I = 1, 2, 3$. Each of these variables is defined as the integral of a (homogeneous) field over the spatial hypersurface. For details, see in particular section V of [6]. We also introduce a massless matter field, described by the variables $\phi$ and $\pi$ (also defined as integrated over space), where $\pi$ is the conjugate momentum. Poisson brackets for the total system read [6]

$$\{P_I, P_J\} = 0 = \{C_I, C_J\}, \qquad \{P_I, C_J\} = 2\delta_J^I C_I, \qquad \{\phi, \pi\} = 1. \tag{1}$$

To connect with the notation that is more common for affine algebras, we perform the partial redefinition of variables $(C_I, P_J) =: (C_I, -2D_J)$, which leads to the affine Poisson brackets [16]

$$\{D_I, D_J\} = 0 = \{C_I, C_J\}, \qquad \{C_J, D_I\} = \delta_J^I C_I. \tag{2}$$

An algebra with such brackets is called an affine Lie algebra.

The dynamics of the system is specified by the equations (We have the extra factor 2 in Equation (3) and (4) that is missing in the corresponding equations of [16].)

$$\dot{D}_I = -C_I(C - 2C_I), \tag{3}$$
$$\dot{C}_I = 4C_I(D - 2D_I), \tag{4}$$
$$\dot{\pi} = 0, \tag{5}$$
$$\dot{\phi} = \kappa\pi, \tag{6}$$

where $D = D_1 + D_2 + D_3$ and $C = C_1 + C_2 + C_3$. There is no summation $\sum_I$ in the rhs of (3) and (4). The solutions to (3)–(6) have to satisfy the Hamiltonian constraint

$$H = \frac{1}{2}C^2 - \sum_I C_I^2 + 4\left(\frac{1}{2}D^2 - \sum_I D_I^2\right) + \frac{\kappa}{2}\pi^2 = 0, \tag{7}$$

where $\kappa = \pm 1$ defines two possible dynamics (two different signatures of the corresponding bilinear forms) with respect to the field $\phi$. Unlike the traditional momentum, which serves to translate the canonical coordinate $C_I$, the variable $D_I$ serves to dilate $C_I$.

The set of equations (3)–(7) incorporates the dynamics of all Bianchi-type A models. It presents a coupled system of nonlinear equations that have not been solved in the general case analytically yet. To get some insight into the local geometry of the space of solutions to these equations, we apply the dynamical systems method [20,21].

It is easy to see that space $S$ of the critical points of the dynamics, defined by the vanishing of the right-hand sides of (3)–(6) and satisfying the constraint (7), reads

$$S = \left\{(C_1, C_2, C_3, D_1, D_2, D_3, \pi, \phi) \in \mathbb{R}^8 \mid (C_I = 0 = \pi) \wedge (D^2 = 2\sum_I D_I^2)\right\}, \tag{8}$$

where $I = 1, 2, 3$ and $D = D_1 + D_2 + D_3$.

The Jacobian of the system (3)–(6) is easily found to be

$$J = \begin{pmatrix} J_{11} & -C_1 & -C_1 & 0 & 0 & 0 & 0 & 0 \\ -C_2 & J_{22} & -C_2 & 0 & 0 & 0 & 0 & 0 \\ -C_3 & -C_3 & J_{33} & 0 & 0 & 0 & 0 & 0 \\ J_{41} & 0 & 0 & -4C_1 & 4C_1 & 4C_1 & 0 & 0 \\ 0 & J_{52} & 0 & 4C_2 & -4C_2 & 4C_2 & 0 & 0 \\ 0 & 0 & J_{63} & 4C_3 & 4C_3 & -4C_3 & 0 & 0 \\ 0 & 0 & 0 & 0 & 0 & 0 & 0 & 0 \\ 0 & 0 & 0 & 0 & 0 & 0 & 0 & \kappa \end{pmatrix}, \tag{9}$$

where

$$J_{11} = 2C_1 - C_2 - C_3, \quad J_{22} = -C_1 + 2C_2 - C_3, \quad J_{33} = -C_1 - C_2 + 2C_3,$$

and where

$$J_{41} = 4(-D_1 + D_2 + D_3), \quad J_{52} = 4(D_1 - D_2 + D_3), \quad J_{63} = 4(D_1 + D_2 - D_3).$$

The Jacobian evaluated at any point of the set (8) is the following matrix

$$
J_S = \begin{pmatrix}
0 & 0 & 0 & 0 & 0 & 0 & 0 & 0 \\
0 & 0 & 0 & 0 & 0 & 0 & 0 & 0 \\
0 & 0 & 0 & 0 & 0 & 0 & 0 & 0 \\
J_{41} & 0 & 0 & 0 & 0 & 0 & 0 & 0 \\
0 & J_{52} & 0 & 0 & 0 & 0 & 0 & 0 \\
0 & 0 & J_{63} & 0 & 0 & 0 & 0 & 0 \\
0 & 0 & 0 & 0 & 0 & 0 & 0 & 0 \\
0 & 0 & 0 & 0 & 0 & 0 & 0 & \kappa
\end{pmatrix}.
\tag{10}
$$

Thus, the characteristic polynomial associated with $J_S$ reads

$$
P(\lambda) = (-\lambda)^7(\kappa - \lambda),
\tag{11}
$$

so that the eigenvalues are the following:

$$
(0, 0, 0, 0, 0, 0, 0, \kappa).
\tag{12}
$$

Since the real parts of all, but one, eigenvalues of the Jacobian $J_S$ are equal to zero, the fixed points defined by Equation (8) are nonhyperbolic (A critical point is called a hyperbolic fixed point if all eigenvalues of the Jacobian matrix of the linearized equations at this point have nonzero real parts. Otherwise, it is called a nonhyperbolic fixed-point [20,21].). Thus, getting insight into the structure of the space of solutions to the dynamics near such points require an examination of the exact form of the dynamics. The information obtained from linearized set of equations is unable to reveal the nature of dynamics in the neighborhood of such fixed points.

In the next subsection, we present explicit, but approximate, solution to our dynamics characterizing the strange spike. It is obtained by solving the dynamics with the so-called inhomogeneous initial data (This definition and example explaining the idea of the inhomogeneous initial data are due to David Sloan.). The latter means that the initial data is not just a set of 3 $C$s and 3 $D$s per point in phase space, but the related data on some curve in this space. For instance, let us choose $(C_1, C_2, C_3) := (\tilde{x}, 0.8, 0.4)$ and $(D_1, D_2, D_3) := (f(\tilde{x}), 2, 7)$, where $f(\tilde{x})$ is the value that solves the Hamiltonian constraint (7) for $C_1 = \tilde{x}$. Next, we allow $\tilde{x}$ to vary from $-0.1$ to $0.1$. Then, instead of solving one set of equations for each point of space, we solve a whole continuous (in practical calculations, discrete) family of them by taking the sequence of $\tilde{x} \in (-0.1, 0.1)$. The plot of the $C$s and the $D$s as functions of $\tilde{x}$ reveals a peculiar structure that emerges in $\tilde{x}$ around $\tilde{x} = 0$ that we call the strange spike (We use $\tilde{x}$ to denote the initial data in phase space sticking to the notation of Section 2 of Ref. [16].)

It results from Equation (5) and (6) that $\phi$ is a monotonic function of time. Thus, it can be used as an evolution parameter of the dynamics. Dividing both sides of (3) and (4) by $\dot{\phi} = \kappa\pi$, we obtain

$$
\kappa\pi \frac{dD_I}{d\phi} = -C_I(C - 2C_I),
\tag{13}
$$

$$
\kappa\pi \frac{dC_I}{d\phi} = 4C_I(D - 2D_I),
\tag{14}
$$

which defines the relative dynamics with respect to the variable $\phi$.

In our case the equations of motion (13)–(14) and the constraint (7) are constructed from elements of the affine algebra (2). This algebra can be realized in terms of the Poisson algebra by the adjoint action

$$
\{X, \cdot\}Y = \{X, Y\},
\tag{15}
$$

where $X$ and $Y$ are linear combinations of basic elements $C_J$ and $D_I$ of affine algebra. Exponentiation of this algebra gives the operators representing the affine group. For fixed $I$ (where $I = 1, 2, 3$) the elements of the affine group in "one direction" are represented by the following operations in the phase space

$$g_I(\alpha_I, \beta_I) = e^{\alpha_I\{C_I,\cdot\}} e^{-\beta_I\{D_I,\cdot\}} \,, \tag{16}$$

where $\alpha_I \in \mathbb{R}$ and $\beta_I \in \mathbb{R}_+$.

The full group is composed of all three independent transformations $g(\alpha_1, \beta_1, \alpha_2, \beta_2, \alpha_3, \beta_3) = g_1(\alpha_1, \beta_1) g_2(\alpha_2, \beta_2) g_3(\alpha_3, \beta_3)$. This property allows, in most cases, to perform calculations for fixed $I$ and to generalize the result to other $I$.

The action of this group on the basic elements of the affine algebra can be summarized as

$$\begin{aligned} g_I(\alpha_I, 1)C_J &= C_J, \quad g_I(\alpha_I, 1)D_J = D_J + \delta_{IJ}\alpha_J C_J \\ g_I(0, \beta_I)C_J &= \delta_{IJ}\beta_J C_J, \quad g_I(\alpha_I, 1)D_J = D_J + \delta_{IJ}\alpha_J C_J. \end{aligned} \tag{17}$$

The manifold of the variables $(C_I, D_I)$ splits into orbits with respect to the affine group. The orbit of the element $(C_{I0}, D_{I0})$ is defined as

$$\Pi^I_{(C_{I0}, D_{I0})} = \{(C_I, D_I) \colon (C_I, D_I) = g_I(\alpha_I, \beta_I)(C_{I0}, D_{I0})\} \,. \tag{18}$$

Using (17), it is easy to check that one gets two large orbits which we denote by $\Pi^I_-$ and $\Pi^I_+$ and continuum of orbits consisting of single points $\Pi^I_{D_I}$

$$\Pi^I_- := \{(C_I, D_I) \mid C_I \in \mathbb{R}_-, D_I \in \mathbb{R}\} \,, \tag{19}$$

$$\Pi^I_+ := \{(C_I, D_I) \mid C_I \in \mathbb{R}_+, D_I \in \mathbb{R}\} \,, \tag{20}$$

$$\Pi^I_{D_I} := \{(0, D_I)\} \text{ where } D_I \in \mathbb{R} \,. \tag{21}$$

In fact, due to the constraint (7) the sets of points in the orbits are reduced to some submanifolds.

Since $C_I = 0$ is a critical point of the system (3)–(4), the sign of each $C_I$ along any dynamical trajectory is fixed by the initial conditions. The orbits $\Pi^I_-$ and $\Pi^I_+$ carry such solutions. Every single-point orbit $\Pi^I_{D_I}$ separates positive ($C_I > 0$) and negative ($C_I < 0$) parts of the kinematical trajectory. However, if the system enters the orbit $\Pi^I_{D_I}$, it is not able to leave it. This property is very strong. It is even fulfilled not only for infinitesimal perturbation of the motion, but also for finite difference form of the equations of motion obtained by the Euler algorithm. Changing in (13)–(14) derivatives into finite differences, one gets

$$\kappa\pi \, D_I(\phi_{n+1}) = -C_I(\phi_n)(C(\phi_n) - 2C_I(\phi_n))\Delta\phi + \kappa\pi \, D_I(\phi_n) \,, \tag{22}$$

$$\kappa\pi \, C_I(\phi_{n+1}) = 4C_I(\phi_n)(D(\phi_n) - 2D_I(\phi_n))\Delta\phi + \kappa\pi \, C_I(\phi_n) \,, \tag{23}$$

where $\phi_{n+1} = \phi_n + \Delta\phi$.

Assuming that for a given $\phi_n$ the point $(C_I(\phi_n) = 0, D_I(\phi_n))$ belongs to the orbit $\Pi^I_{D_I}$, the resulting point $(C_I(\phi_{n+1}) = 0, D_I(\phi_{n+1}) = D_I(\phi_n))$ also belongs to the same orbit $\Pi^I_{D_I}$ independently of $\Delta\phi$.

These single-point orbits separate classical trajectories into $C_I > 0$ and $C_I < 0$ regions. In fact, the region to which belong given trajectory depends on sign of $C_I(\phi_0)$ of the initial conditions $(C_I(\phi_0) = 0, D_I(\phi_0))$. This is shown in the next section.

Space $S$ (see (8)) consists of the nonhyperbolic critical points so that the neighborhood of each such point includes rapidly changing trajectories. Thus, the trajectories approaching asymptotically the orbits $\Pi^I_{D_I}$ are very sensitive to the choice of the initial data for the dynamics. This can be seen in Figures 1 and 2 (for the case $I = 1$). These neighborhoods represent the strange spikes.

It is expected that quantization may smear out the regions around the single-point orbits so that the corresponding spikes may become more smooth.

### 2.2. Classical Spikes

### 2.2.1. Parametrization of Dynamics by a Scalar Field

In order to derive the spike solutions, within the dynamics parameterized by the scalar field $\phi$, we follow the approach presented in Section 2 of Ref. [16].

Let us assume that the initial conditions for $D_I$ and $C_I$ at $\phi = \phi_0$ have the form: $D_1 < D_2 < D_3 < 0$ and $1 \gg C_I > 0$. Then, it follows from (13)–(14) that $C_2$ and $C_3$ almost instantly vanish, while $D_2$ and $D_3$ turn out to be essentially constant. For later convenience we define $D_\pm := D_2 + D_3 \pm 2\sqrt{D_2 D_3}$. Therefore, the problem reduces to finding the evolution of $C_1$ and $D_1$, which is governed by the equations (here we denote by prime the derivative with respect to $\phi$):

$$\kappa \pi C_1' = 4C_1(-D_1 + D_2 + D_3)\,, \tag{24}$$

$$\kappa \pi D_1' = C_1^2\,, \tag{25}$$

$$-C_1^2 = 4(D_1 - D_+)(D_1 - D_-) - \kappa \pi^2\,, \tag{26}$$

where the last one results from the constraint (7). Inserting the right-hand side of (26) into (25), we obtain an equation independent of $C_1$, whose solution can be written as

$$\begin{aligned}
D_1(\phi) = D_2 + D_3 + \frac{1}{2}\sqrt{16 D_2 D_3 + \kappa \pi^2}\ \tanh\bigg(&\frac{2}{\kappa \pi}\sqrt{16 D_2 D_3 + \kappa \pi^2}\,(\phi - \phi_0) \\
&-\operatorname{arctanh}\sqrt{\frac{16 D_2 D_3 - C_{10}^2 + \kappa \pi^2}{16 D_2 D_3 + \kappa \pi^2}}\,\bigg).
\end{aligned} \tag{27}$$

In the above expression the initial condition $D_1(\phi_0) = D_{10}$ has been replaced by

$$D_1(\phi_0) := D_{10} = D_2 + D_3 - \frac{1}{2}\sqrt{16 D_2 D_3 - C_{10}^2 + \kappa \pi^2}\,, \tag{28}$$

due to the relation (26) for $C_1(\phi_0) = C_{10}$. Furthermore, (27) and (26) give

$$\begin{aligned}
C_1(\phi) = \operatorname{sgn}(C_{10})\sqrt{16 D_2 D_3 + \kappa \pi^2}\ \operatorname{sech}\bigg(&\frac{2}{\kappa \pi}\sqrt{16 D_2 D_3 + \kappa \pi^2}\,(\phi - \phi_0) \\
&-\operatorname{arctanh}\sqrt{\frac{16 D_2 D_3 - C_{10}^2 + \kappa \pi^2}{16 D_2 D_3 + \kappa \pi^2}}\,\bigg),
\end{aligned} \tag{29}$$

where "sgn" denotes the sign function (its value for $C_{10} = 0$ is irrelevant since then $C_1(\phi) = 0$). One can verify that (29) together with (27) solve the equations (24) and (26).

Choosing the simple parametrization $C_{10} = \tilde{x}$, we can now draw $D_1$ and $C_1$ as functions of both the evolution parameter $\phi$ and the initial condition $\tilde{x}$, or as functions of only $\tilde{x}$, for different fixed values of $\phi$. Figures 1 and 2 present the corresponding plots for the setting of other quantities: $D_2 = -2$, $D_3 = -1$, $\kappa = 1$, $\pi = 1$ and $\phi_0 = 0$. One can see that $D_1(\phi)$ and $C_1(\phi)$ behave in the same way as $P_1(t)$ and $C_1(t)$ presented in Ref. [16], which is expected as the evolution parameter $\phi$ is a monotonic function of the evolution parameter $t$ owing to Equations (5) and (6).

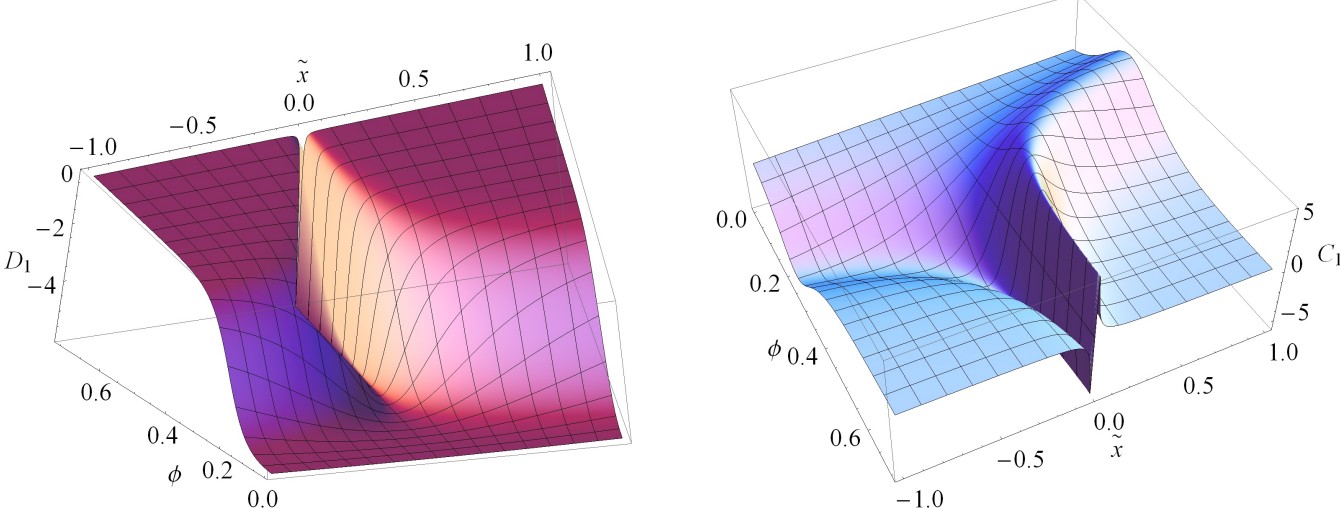

**Figure 1.** $D_1(\tilde{x}, \phi)$ (left) and $C_1(\tilde{x}, \phi)$ (right) as functions of two variables; lines of constant $\tilde{x}$ correspond to orbits (19)–(21) of the affine group

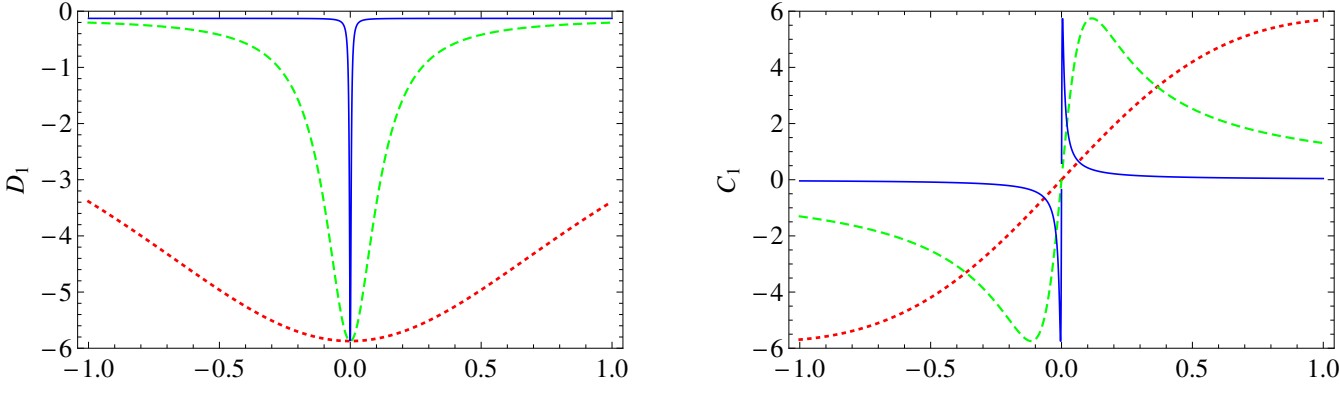

**Figure 2.** $D_1(\tilde{x}, \phi)$ (left) and $C_1(\tilde{x}, \phi)$ (right) as functions of $\tilde{x}$ for fixed values of $\phi = 0.2$ (red, dotted), $\phi = 0.4$ (green, dashed) and $\phi = 0.7$ (blue, solid); these are cross-sections of plots from Figure 1.

### 2.2.2. Parametrization of Dynamics by the Arc Length

The arc length of the curve $\vec{r}(\tilde{x}) \equiv (C_1(\tilde{x}), D_1(\tilde{x}))$ is given by the integral

$$s(\tilde{x}) = \int_{\tilde{x}_0}^{\tilde{x}} dy \sqrt{\left(\frac{dC_1(y)}{dy}\right)^2 + \left(\frac{dD_1(y)}{dy}\right)^2}, \tag{30}$$

where $\tilde{x}_0$ is a certain chosen minimal value of $\tilde{x}$. Calculating (30) we obtain

$$s(x) = -\frac{1}{2}\sqrt{16D_2D_3 + \kappa\pi^2}\left(i\,\mathrm{E}(i\,\zeta, 4) - i\,\mathrm{F}(i\,\zeta, 4) + \sqrt{2\cosh(2\zeta) - 1}\,\tanh\zeta\right),$$

$$\zeta \equiv \frac{2}{\kappa\pi}\sqrt{16D_2D_3 + \kappa\pi^2}\,(\phi - \phi_0) - \mathrm{arctanh}\frac{\tilde{x}}{\sqrt{16D_2D_3 + \kappa\pi^2}}, \tag{31}$$

where F denotes the elliptic integral of the first kind and E of the second kind. This allows us to express the curve $\vec{r}(\tilde{x})$ as a function of $s$, which needs to be calculated numerically. Introducing the (normalized) Frenet vectors

$$\hat{e}_1(s) := \frac{1}{|\vec{e}_1(s)|}\,\vec{r}'(s), \qquad \hat{e}_2(s) := \frac{1}{|\vec{e}_2(s)|}\left(\vec{r}''(s) - \vec{r}''(s) \cdot \vec{e}_1(s)\,\vec{e}_1(s)\right), \tag{32}$$

one can define the generalized curvature of $\vec{r}(s)$ as follows (see, e.g., [22])

$$\chi(s) := \frac{1}{|\vec{r}'(s)|} \hat{e}_1'(s) \cdot \hat{e}_2(s) . \tag{33}$$

In Figure 3 we depict the generalized curvature of the curve $(C_1(s), D_1(s))$ as a function of the normalized arclength $\bar{s}$ corresponding to $\tilde{x} \in [-5, 5]$ (i.e., $s$ divided by the maximal value $s(\tilde{x} = 5)$, for a given $\phi$) for different values of the evolution parameter $\phi$. The values of $\kappa$, $\pi$, $\phi_0$ and $D_2$, $D_3$ are kept the same as in the previous subsection. Moreover, dots on the horizontal axis denote the value of $\bar{s}(\tilde{x} = 0)$ for a given $\phi$, which naturally coincides with the middle of the spike. The double peak corresponds to the two inflection points of the curve visible on the right plot in Figure 2.

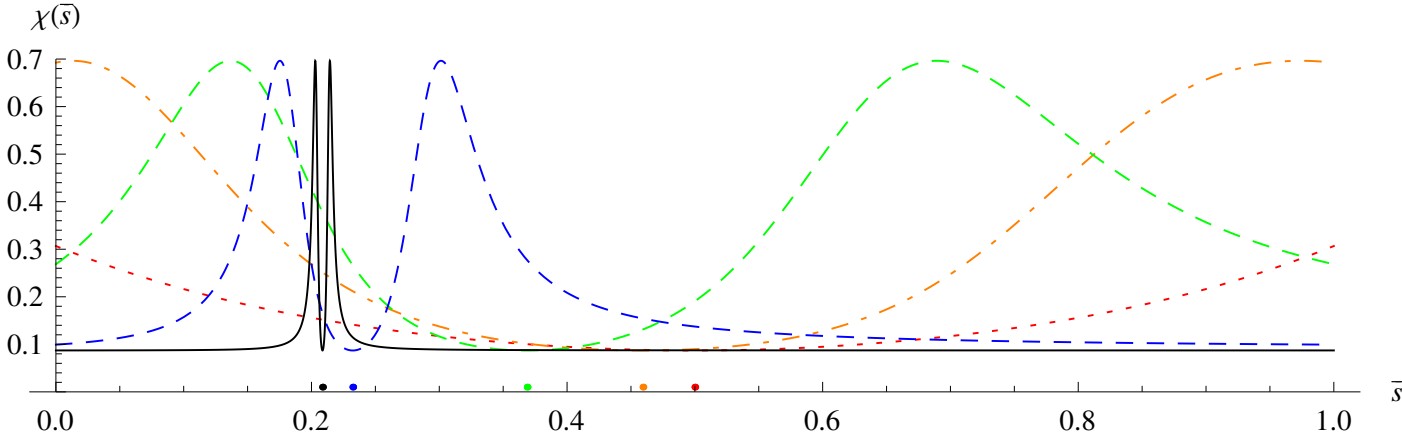

**Figure 3.** The generalized curvature $\chi(\bar{s})$ of $(C_1(\bar{s}), D_1(\bar{s}))$ for evolution parameters $\phi = 0$ (red, dotted), $\phi = 0.05$ (orange, dot-dashed), $\phi = 0.1$ (light green, dashed), $\phi = 0.2$ (dark blue, dashed), and $\phi = 0.4$ (black, solid).

Figure 3 shows that the spike is created at some moment in the evolution of the gravitational system and seems to be permanent. The shape of the spike depends on time and changes from a plateau to a singular structure.

## 3. Quantum Level

### 3.1. Representation of the Affine Group

The quantum version of the Lie algebra (2) is defined by the algebraic quantization principle: $C_I \to \hat{C}_I$ and $D_I \to \hat{D}_I$, such that (Throughout the paper we choose $\hbar = 1$ and use Planck's units except where otherwise stated.)

$$[\hat{C}_I, \hat{C}_J] = 0 = [\hat{D}_I, \hat{D}_J], \quad [\hat{C}_J, \hat{D}_I] = i\, \delta_J^I \hat{C}_I . \tag{34}$$

where $I, J = 1, 2, 3$. The commutation relations (34) are the same as for the generators of the affine group [23].

The affine group $\text{Aff}(\mathbb{R}_+)_I$ generated by the pair $\hat{C}_I$ and $\hat{D}_I$ has two inequivalent unitary representations $U_-(p, q)_I$ and $U_+(p, q)_I$. They are constructed in two carrier spaces of square-integrable functions $L^2(\mathbb{R}_-, d\nu(x^I))$ and $L^2(\mathbb{R}_+, d\nu(x^I))$, $d\nu(x^I) = dx^I/|x^I|$, which correspond to the negative and positive spectrum of the position operator $\hat{C}_I$, respectively. Because of physical interpretation we needs the full spectrum of the position operator. This requirement enforces using the reducible representation of the affine group in the carrier space $\mathcal{K}_I := L^2(\mathbb{R}_-, d\nu(x^I)) \oplus L^2(\mathbb{R}_+, d\nu(x^I))$. The general form of the vector $f \in \mathcal{K}_I$ can be written as a direct sum of the functions $f_\mp \in L^2(\mathbb{R}_\mp, d\nu(x^I))$:

$$f = f_- \oplus f_+ . \tag{35}$$

The scalar product of such two vectors is the sum of the appropriate partial scalar products:

$$\langle f_1 \oplus f_2 | g_1 \oplus g_2 \rangle := \langle f_1 | g_1 \rangle_- + \langle f_2 | g_2 \rangle_+$$
$$= \int_{-\infty}^{0} d\nu(x^I) f_1(x^I)^\star g_1(x^I) + \int_{0}^{\infty} d\nu(x^I) f_2(x^I)^\star g_2(x^I) . \qquad (36)$$

The action of the affine group $\mathrm{Aff}(\mathbb{R}_+)_I$ in this carrier space $\mathcal{K}_I$ can be written as

$$U(p,q)_I f = U_-(p,q)_I f_- \oplus U_+(p,q)_I f_+ , \qquad (37)$$

where $p \in \mathbb{R}$, $q \in \mathbb{R}_+$ and

$$U_\mp(p,q)_I f_\mp(x^I) = e^{ipx^I} f_\mp(qx^I) . \qquad (38)$$

This structure allows for extension of this affine action to the whole straight line. For this purpose it is enough to extend the appropriate functions from half-line to the full straight line: $f_-(x^I) = 0$ for $x^I \geq 0$ and $f_+(x^I) = 0$ for $x^I \leq 0$. Then, denoting by $|x^I \oplus x^I\rangle$ the "position" vector in the space $\mathcal{K}_I$, every function belonging to $\mathcal{K}_I$ can be represented as:

$$f(x^I) := \langle x^I \oplus x^I | f_- \oplus f_+ \rangle = \langle x^I | f_- \rangle + \langle x^I | f_+ \rangle = f_-(x^I) + f_+(x^I). \qquad (39)$$

It is obvious that the space $\mathcal{K}_I \subset L^2(\mathbb{R}, d\nu(x^I))$ and that the scalar product (36) can be rewritten as

$$\langle f_1 \oplus f_2 | g_1 \oplus g_2 \rangle_I = \langle f | g \rangle_I = \int_{-\infty}^{\infty} d\nu(x^I) f(x^I)^\star g(x^I) . \qquad (40)$$

The action of the affine group $\mathrm{Aff}(\mathbb{R}_+)_I$ in this new carrier space, which we denote again by $\mathcal{K}_I$, can be written as

$$U(p,q)_I f(x^I) = e^{ipx^I} f(qx^I) . \qquad (41)$$

The explicit representation of the generators of this group are given by the following operators

$$\hat{D}_I f(x^I) := -i\, x^I \frac{\partial}{\partial x^I} f(x^I), \qquad \hat{C}_I f(x^I) := x^I f(x^I), \qquad (42)$$

where $I = 1, 2, 3$.

The corresponding unitary operators representing elements of the affine group are:

$$\hat{U}(p,q)_I = e^{ip\hat{C}_I} e^{i\ln(q)\hat{D}_I} \qquad (43)$$

where $-\infty < p < +\infty$, $0 < q < +\infty$.

Taking into account three variables $x^I$ $(I = 1, 2, 3)$, the carrier space $\mathcal{K}$ for the representation of the algebra (34) can be defined to be

$$\mathcal{K} := \mathcal{K}_1 \otimes \mathcal{K}_2 \otimes \mathcal{K}_3 , \qquad (44)$$

where

$$\mathcal{K}_I = L^2(\mathbb{R}_-, d\nu(x^I)) \oplus L^2(\mathbb{R}_+, d\nu(x^I)) \subset L^2(\mathbb{R}, d\nu(x^I)) \qquad (45)$$

and the scalar product is constructed according to prescription for tensor product of Hilbert spaces:

$$\langle f | g \rangle = \int_{-\infty}^{\infty} d\nu(x^1) \int_{-\infty}^{\infty} d\nu(x^2) \int_{-\infty}^{\infty} d\nu(x^3) f(x^1, x^2, x^3)^\star g(x^1, x^2, x^3) . \qquad (46)$$

The "total" affine group used in this paper is the direct product of the three affine groups $\mathrm{Aff}_0 = \mathrm{Aff}(\mathbb{R}_+)_1 \otimes \mathrm{Aff}(\mathbb{R}_+)_2 \otimes \mathrm{Aff}(\mathbb{R}_+)_3$. This realization of the affine group allows for the physical interpretation of quantized $C_I$ and $D_I$ variables.

*3.2. Quantum Dynamics*

The quantum dynamics of our system may be derived, to some extent, from the quantum version of the Hamiltonian constraint defined by Equation (7). In a standard approach, one maps the dynamical constraint into an operator defined in kinematical Hilbert space. Its kernel may be used to construct physical Hilbert space. However, such an approach leads to the problem of time at the quantum level.

The reason for having the scalar field in the Hamiltonian (7), is the hope that it may resolve the problem of time both at classical and quantum levels. Such an approach works in the classical case as it leads to the relative dynamics, defined by Equations (13)–(14), parameterized by the scalar field $\phi$. However, an extension of this strategy to the quantum level faces serious difficulty. Namely, quantization of the scalar field algebra $\{\phi, \pi\} = 1$ as follows

$$\hat{\pi}f(\phi) := -i\frac{\partial}{\partial\phi}f(\phi), \quad \hat{\phi}f(\phi) := \phi f(\phi), \quad f \in L^2(\mathbb{R}, d\phi), \tag{47}$$

so that $[\hat{\phi}, \hat{\pi}] = i\mathbb{I}$, leads to the inconsistency. In this case, according to the standard approach to quantum mechanics, $\phi$ represents an additional degree of freedom of our quantum system. The field $\phi$ is the variable involved in every required wave function and it cannot be considered as a parameter representing a reference clock we want to introduce. One needs to notice that every quantum amplitudes is independent of the variable $\phi$ because they are obtained by calculating the appropriate scalar product containing among others integration over $\phi$.

To parameterize with $\phi$ the reference classical clock uncoupled to our quantum system one needs to construct a hybrid approximation of deterministic unitary quantum evolution: the field should evolve in a classical way and quantum states of the system should evolve according to a unitary prescription.

Let us treat the field $\phi$ as a classical field which value is considered as a parameter showing tics of a classical clock, that is $\phi$ is a parameter enumerating changes of our Hamiltonian system.

We propose to modify Schrödinger type unitary evolution operator to the form containing both: evolution of the classical field and evolution of the quantum system itself. This operator we denote by $\mathcal{U}(\phi, \phi_0)$. It is defined by a series of natural conditions:

- First of all, the operator $\mathcal{U}(\phi, \phi_0)$ evolves the quantum state of our gravitational system from the "time" $\phi_0$ and the state $\Psi_1$ to the "time" $\phi$ and the state $\Psi_2$ as follows

$$\mathcal{U}(\phi, \phi_0)\Psi_1(\phi_0, x_1, x_2, x_3) = \Psi_2(\phi, x_1, x_2, x_3), \tag{48}$$

where $\Psi_1, \Psi_2 \in \mathcal{K}$, and where $\mathcal{K}$ is a Hilbert space. Thus $\mathcal{U}$ changes the state vector in the Hilbert state space and the time parameter by changing the field.

Since the time parameter $\phi$ does not couple to the gravitational field in (7), we can factorize the evolution operator $\mathcal{U}(\phi, \phi_0)$ into two independent operations:

(a) the unitary operator $V_{\mathcal{K}}(\phi, \phi_0)$ acting on the spatial dependence of state vectors in the Hilbert space $\mathcal{K}$ while the field $\phi$ is changing,

(b) the operation $V_{\pi}(\phi, \phi_0)$ acting on the parametric dependence of the state vectors of the field $\phi$.

In what follows, we assume the dependence of the evolution operator on the difference $\tau = \phi - \phi_0$ between the final and initial value of the field $\phi$, i.e., we assume the translational invariance of the evolution operator with respect to the parameter $\phi$. This means that $\mathcal{U}$ does not depend on the choice of the initial time $\phi_0$, but only on $\tau$. Thus, the full evolution operator can be written as

$$\mathcal{U}(\tau) = V_{\mathcal{K}}(\tau)V_{\pi}(\tau). \tag{49}$$

- The evolution operator fulfils the standard conditions for quantum evolution:

$$\mathcal{U}(0) = \hat{\mathbb{1}} \quad \text{(no shift in ''time'')} \, , \tag{50}$$

$$\mathcal{U}(\tau_2 + \tau_1) = \mathcal{U}(\tau_2)\mathcal{U}(\tau_1) \quad \text{(no ''holes'' in the evolution)} \, , \tag{51}$$

$$\mathcal{U}(\tau)^{\dagger} = \mathcal{U}(\tau)^{-1} = \mathcal{U}(-\tau) \quad \text{(unitarity)} \, . \tag{52}$$

The first one represents the fact, that if there is no shift in time, the state vector stays the same. The second means that every evolution can be split into intermediate steps. These two conditions are expected to hold for both the classical end quantum evolution. The last line represents the unitarity condition which is related to the probabilistic interpretation of quantum mechanics.

To fulfil the last condition the parametric part of the evolution operator has to transform as the complex conjugation:

$$[V_{\mathcal{K}}(\tau)V_{\pi}(\tau)]^{\dagger} = V_{\mathcal{K}}(\tau)^{\dagger}V_{\pi}(\tau)^{\star} \tag{53}$$

Let us now consider a formal shift operation with respect to the field $\phi$. For this purpose, we define a kind of adjoint action of the field $\phi$ and its canonically conjugate momentum $\pi$ on the classical phase space. For an arbitrary function $g(\phi, \pi)$ on this phase space the adjoint action is defined to be

$$\begin{aligned} \{h(\phi, \pi), \cdot\} f(\phi, \pi) &:= \{h(\phi, \pi), f(\phi, \pi)\} \, , \\ \{\cdot, h(\phi, \pi)\} f(\phi, \pi) &:= \{f(\phi, \pi), h(\phi, \pi)\} \, , \\ \{h(\phi, \pi), \cdot\} &= -\{\cdot, h(\phi, \pi)\} \, , \end{aligned} \tag{54}$$

where the Poisson bracket is given by

$$\{h(\phi, \pi), f(\phi, \pi)\} := \frac{\partial h(\phi, \pi)}{\partial \phi} \frac{\partial f(\phi, \pi)}{\partial \pi} - \frac{\partial h(\phi, \pi)}{\partial \pi} \frac{\partial f(\phi, \pi)}{\partial \phi} \, . \tag{55}$$

One can directly check that

$$e^{\tau\{\cdot, \pi\}} f(\phi, \pi) = e^{-\tau\{\pi, \cdot\}} f(\phi, \pi) = f(\phi + \tau, \pi) \, , \tag{56}$$

where

$$e^{\tau\{\pi, \cdot\}} = \sum_{n=0}^{\infty} \frac{\tau^n \{\pi, \cdot\}^{(n)}}{n!} \, . \tag{57}$$

The powers of the adjoint action are understood as

$$\{\pi, \cdot\}^{(n)} f(\phi, \pi) = \underbrace{\{\pi, \{\pi, \dots \{\pi, \{}_{n} \pi, f(\phi, \pi)\} \dots\} \, , \tag{58}$$

where $\{\pi, \cdot\}^{(0)} f(\phi, \pi) = f(\phi, \pi)$.

The isomorphic realization of the classical shift operation (56) with respect to the field $\phi$ in the state space is given by

$$e^{\tau\{\pi, \cdot\}} \to e^{\tau\frac{\partial}{\partial \phi}} \, . \tag{59}$$

As a consequence the formal shift of the state $\Psi(\phi, x)$ in respect to the time $\phi$ is given by

$$\Psi(\phi + \tau, x) = e^{\tau\frac{\partial}{\partial \phi}} \Psi(\phi, x) \, , \tag{60}$$

where $x := (x_1, x_2, x_3)$.

The comparison of the operations (60) and (56) suggests that the shift generator $\frac{\partial}{\partial \phi} =: \breve{\pi}$, defined in the quantum state space, may play a similar role to the classical

momentum $\pi$ acting (by the adjoint action) in the phase space. Working in the quantum state space we postulate the replacement of the classical momentum $\pi$ with the operation $\check{\pi}$.

The classical evolution in the phase space can be written in terms of the adjoint action as $e^{\tau\{E(\pi),\cdot\}}$, where $E(\pi)$ is a real function of the momentum $\pi$ generating evolution of this free field. As a consequence of (59) the appropriate realization of this operation to be a part of the evolution operator is the replacement $E(\pi) \to -iE(\check{\pi})$. The imaginary unit has to be added to fulfil the unitarity requirement (53). The classical part of the evolution operator is expected to be:

$$V_\pi(\tau) = e^{-i\tau E(\check{\pi})}, \tag{61}$$

where $E(\check{\pi})$ is some real function of $\check{\pi}$.

Making use of (60)–(61) and the factorization (49), we rewrite (48) as follows

$$e^{\tau\check{\pi}}\Psi(\phi,x) = V_\mathcal{K}(\tau)e^{-i\tau E(\check{\pi})}\Psi(\phi,x) \tag{62}$$

Taking derivative of (62) with respect to $\tau$, at $\tau = 0$, leads to the local evolution equation:

$$\check{\pi}\Psi(\phi,x) = \left[\left(\frac{\partial V_\mathcal{K}(\tau)}{\partial\tau}\right)_{\tau=0} - iE(\check{\pi})\right]\Psi(\phi,x). \tag{63}$$

Introducing

$$\hat{W} := i\left(\frac{\partial V_\mathcal{K}(\tau)}{\partial\tau}\right)_{\tau=0}, \tag{64}$$

we can rewrite (63) in the form

$$i\frac{\partial\Psi(\phi,x)}{\partial\phi} = \left[\hat{W} + E\left(\frac{\partial}{\partial\phi}\right)\right]\Psi(\phi,x). \tag{65}$$

Assuming

$$\Psi(\phi,x) = \omega(\phi)\psi(x), \tag{66}$$

enables rewriting (65) in the separable form

$$\frac{1}{\omega(\phi)}\left[i\frac{\partial}{\partial\phi} - E\left(\frac{\partial}{\partial\phi}\right)\right]\omega(\phi) = \frac{1}{\psi(x)}\hat{W}\psi(x), \tag{67}$$

which leads to the two eigenequations:

$$\left[i\frac{\partial}{\partial\phi} - E\left(\frac{\partial}{\partial\phi}\right)\right]\omega_\lambda(\phi) = \lambda\omega_\lambda(\phi), \tag{68}$$

and

$$\hat{W}\psi_\lambda(x) = \lambda\psi_\lambda(x). \tag{69}$$

We assume, according to (49), that the quantum evolution operator corresponding to the classical constraint consists of the quantized position-dependent part of (7) and the shifted parametric part of (7). This way we avoid quantization of the algebra $\{\phi,\pi\} = 1$, and consequently quantization of the classical time variable $\phi$. Both classical and quantum evolutions are now parameterized by a single variable $\phi$ that we call the time.

Since there are no products of $C_I$ and $D_I$ in (7), and due to (34), the mapping of $H$ defined by (7) into a Hamiltonian operator $\hat{H}$ is straightforward. We get

$$\begin{aligned} \hat{H} &= E\left(\frac{\partial}{\partial\phi}\right) + \hat{W} \\ &= 2\left(-\frac{\kappa}{4}\frac{\partial^2}{\partial\phi^2} + \sum_I x_I^2\frac{\partial^2}{\partial x_I^2} - 2\sum_{I<J} x_I x_J\frac{\partial^2}{\partial x_I\partial x_J} + \sum_I x_I\frac{\partial}{\partial x_I}\right) + \sum_{I<J} x_I x_J - \frac{1}{2}\sum_I x_I^2, \end{aligned} \tag{70}$$

which implies that

$$\hat{W} = 2\left(\sum_I x_I^2 \frac{\partial^2}{\partial x_I^2} - 2\sum_{I<J} x_I x_J \frac{\partial^2}{\partial x_I \partial x_J} + \sum_I x_I \frac{\partial}{\partial x_I}\right) + \sum_{I<J} x_I x_J - \frac{1}{2}\sum_I x_I^2, \quad (71)$$

$$E\left(\frac{\partial}{\partial \phi}\right) = -\frac{\kappa}{2}\frac{\partial^2}{\partial \phi^2}. \quad (72)$$

3.2.1. Solving the Eigenequation (68) Analytically

Making use of (72), we get (68) in the form

$$\left(i\frac{d}{d\phi} + \frac{\kappa}{2}\frac{d^2}{d\phi^2}\right)\omega_\lambda(\phi) = \lambda\omega_\lambda(\phi). \quad (73)$$

The solution to (73), for $\kappa\lambda \neq 1/2$, is found to be

$$\omega_\lambda(\phi) = e^{-i\kappa\phi}\{A_\lambda \exp(\kappa\sqrt{2\kappa\lambda-1}\,\phi) + B_\lambda \exp(-\kappa\sqrt{2\kappa\lambda-1}\,\phi)\}, \quad (74)$$

whereas for $\kappa\lambda = 1/2$ one has

$$\omega_\lambda(\phi) = (A_\lambda\phi + B_\lambda)e^{-i\kappa\phi}, \quad (75)$$

where $A_\lambda$ and $B_\lambda$ are arbitrary constants. In what follows we denote the solutions (74)–(75) as $\omega_\lambda(A_\lambda, B_\lambda; \phi)$.

3.2.2. Solving the Eigenequation (69) by Variational Method

The eigenequation (69) can be solved numerically in terms of given finite basis of functions $\{\psi_n\}_{n=0}^N$, by taking the solution $\psi_\lambda$ in the form

$$\psi_\lambda \simeq \psi_\lambda^N = \sum_{n=0}^N c_n\psi_n, \quad (76)$$

where $c_n$ are unknown coefficients to be determined. The functions $\psi_n$ should be consistent with the boundary conditions. It means, they should vanish sufficiently fast at zero and infinity to satisfy the condition

$$\|\psi_\lambda\|^2 = \int_{-\infty}^\infty \frac{dx_1}{|x_1|}\int_{-\infty}^\infty \frac{dx_2}{|x_2|}\int_{-\infty}^\infty \frac{dx_3}{|x_3|}|\psi_\lambda|^2 < \infty. \quad (77)$$

The coefficients $c_n$ can be found by considering the following functional:

$$R[\psi_\lambda] := \frac{\|\hat{W}\psi_\lambda - \lambda\psi_\lambda\|^2}{\|\psi_\lambda\|^2}. \quad (78)$$

It is clear that (78) vanishes identically if $\psi_\lambda^N$ is an exact solution to the Equation (69). If this is not the case but $R[\psi_\lambda] \ll 1$, then we have an approximate solution. The smaller $R[\psi_\lambda]$, the better the approximation. The latter fact suggests a method of finding the numerical solution. Namely, one can minimize (78) with respect to all unknown coefficients, including the eigenvalue $\lambda$. This fixes all the parameters in Equation (76) and determines the error $R[\psi_\lambda]$.

To start the procedure one should fix the basis $\{\psi_n\}$. It is reasonable to incorporate the fact that the operator $\hat{W}$ is invariant under $S_3$ group of permutations of the variables $\{x_1, x_2, x_3\}$. Therefore, when looking for the basis, it is reasonable to consider functions

sharing this symmetry, i.e., requiring they are symmetric with respect to the replacements $x_i \leftrightarrow x_j$. A convenient choice is provided by the following ansatz

$$(\psi_S)_\alpha^N = |x_1 x_2 x_3|^\alpha \sum_{n_1 + n_2 + n_3 \le N} \frac{c_{(n_1 n_2 n_3)}}{(n_1 + n_2 + n_3)!} \left( \ln x_1^2 \right)^{n_1} \left( \ln x_2^2 \right)^{n_2} \left( \ln x_3^2 \right)^{n_3} \cdot$$
$$\cdot \exp\left( -\frac{1}{2}(\gamma + i\tilde{\gamma})(|x_1| + |x_2| + |x_3|) \right), \tag{79}$$

where $\alpha \ge \frac{1}{2}$, $\gamma > 0$, $\tilde{\gamma} \in \mathbb{R}$, while $\sum_{n_1 + n_2 + n_3 \le N}$ stands for the sum over $n_1, n_2, n_3 \in [0, N]$ such that $n_1 + n_2 + n_3 \le N$, i.e., the series (79) is terminated at the $N$-th order ($N = n_1 + n_2 + n_3$). The bracket $(n_1, n_2, n_3)$ denotes ordering operation, e.g., $c_{(023)} = c_{023}$, $c_{(203)} = c_{023}$, etc. The operation guaranties that the function $\psi_\lambda^N$ consists of symmetric terms with respect to the replacement $x_i \leftrightarrow x_j$. For instance, there are two second-order ($N = 2$) terms in (79): $\frac{1}{2} c_{011} (\ln x_1^2 \ln x_2^2 + \ln x_1^2 \ln x_3^2 + \ln x_2^2 \ln x_3^2)$ and $\frac{1}{2} c_{002} \left( (\ln x_1^2)^2 + (\ln x_2^2)^2 + (\ln x_3^2)^2 \right)$. The additional weights $1/(n_1 + n_2 + n_3)!$ are introduced for technical simplicity. They guarantee that the coefficients $c_{n_1 n_2 n_3}$, fixed by the minimization procedure, are of a similar order. The latter improve the minimization. Note that the number of $c_{n_1 n_2 n_3}$ grows fast with order $N$.

Taking $\alpha = \frac{1}{2}$ and $N = 10$, one finds a solution $\psi_{\lambda_1} = (\psi_S)_{1/2}^{10}$ specified by numerical parameters $(\lambda, \gamma, \tilde{\gamma}, R[\psi_{\lambda_1}]) = (\lambda_1, \gamma_1, \tilde{\gamma}_1, R_1)$, where

$$\lambda_1 \simeq -0.0821, \quad \gamma_1 \simeq 1.229, \quad \tilde{\gamma}_1 = -4.05 \times 10^{-4} \quad R_1 \simeq 0.0172, \tag{80}$$

while the coefficients $c_{n_1 n_2 n_3}$ are given explicitly in Appendix A. The choice $\alpha = \frac{1}{2}$ leads to the smallest global error $R_1$. The point-like precision defined as

$$E[\psi_\lambda] := \sup_{x_I \in \mathbb{R}^3} \left( |\hat{W}\psi_\lambda - \lambda\psi_\lambda| \right) \tag{81}$$

gives $E[\psi_{\lambda_1}] \simeq 0.0028$. In (81) $\psi_\lambda$ stands for a normalized function.

For the second numerical solution we take the ansatz:

$$(\psi_A)_\alpha^N = (\text{sign}(x_1) + \text{sign}(x_2) + \text{sign}(x_3))(\psi_S)_\alpha^N. \tag{82}$$

As the solution is antisymmetric, it is orthogonal to the previous one, i.e., $\langle (\psi_A)_{\alpha_1}^{N_1} | (\psi_S)_{\alpha_2}^{N_2} \rangle = 0$. Taking as before $\alpha = \frac{1}{2}$ and $N = 10$, we get the function $\psi_{\lambda_2} = (\psi_A)_{1/2}^{10}$. Applying our method of fixing the coefficients in the ansatz leads to

$$\lambda_2 \simeq -0.0957, \quad \gamma_2 \simeq 1.369, \quad \tilde{\gamma}_2 \simeq 4.46 \times 10^{-3}, \quad R_2 \simeq 0.0218. \tag{83}$$

The coefficients $c_{n_1 n_2 n_3}$ are listed in Appendix A. As in the case of $\psi_{\lambda_1}$ the point-like precision (81) gives $E[\psi_{\lambda_2}] \simeq 0.0058$.

### 3.2.3. Solving the Eigenequation (69) by Spectral Method

We start with the ansatz

$$\psi = |x_1 x_2 x_3|^\alpha f(x_1, x_2, x_3) \exp\left( -\frac{\gamma}{2}(|x_1| + |x_2| + |x_3|) \right), \tag{84}$$

where $\gamma > 0$, $\alpha \ge 1/2$. The eigenequation (69) can be rewritten as

$$\hat{W}\psi - \lambda\psi = |x_1 x_2 x_3|^\alpha \left( \hat{F}_{\alpha\gamma}(f(x_1, x_2, x_3)) - \lambda f(x_1, x_2, x_3) \right) \exp\left( -\frac{\gamma}{2}(|x_1| + |x_2| + |x_3|) \right), \tag{85}$$

where

$$\hat{F}_{\alpha\gamma} = 2 \sum_I x_I^2 \frac{\partial^2}{\partial x_I^2} - 4 \sum_{I<J} x_I x_J \frac{\partial^2}{\partial x_I \partial x_J}$$
$$+ 2 \sum_I (1 - 2\alpha + \gamma(|x_1| + |x_2| + |x_3| - 2|x_I|)) x_I \frac{\partial}{\partial x_I} \tag{86}$$
$$+ \sum_I \left( \frac{\gamma^2 - 1}{2} x_I^2 + \gamma(2\alpha - 1)|x_I| \right) + \sum_{I<J} \left( x_I x_J + \gamma^2 |x_I||x_J| \right) - 6\alpha^2.$$

With the ansatz (84), the problem of solving the eigenequation (69) reduces to the problem of solving the corresponding eigenequation for $\hat{F}$ operator, i.e.,

$$\hat{F}_{\alpha\gamma} f(x_1, x_2, x_3) = \lambda f(x_1, x_2, x_3). \tag{87}$$

We solve Equation (87) by using the spectral methods [24]. This form is much more convenient from the numerical point of view because of the lack of two terms $|x_1 x_2 x_3|^\alpha$ and $\exp(-\gamma/2(|x_1| + |x_2| + |x_3|))$, causing additional numerical errors (More precisely, within the spectral method, these terms result in the combination of small and large numbers (components of the matrix representing an approximate form of the eigenequation at a lattice).).

It is convenient, for numerical treatment, to assume

$$f(x_1, x_2, x_3) = \sum_{n_1=1}^{N} \sum_{n_2=1}^{N} \sum_{n_3=1}^{N} c_{n_1,n_2,n_3} f_{n_1 n_2 n_3}(x_1, x_2, x_3), \tag{88}$$

where $N > 1$ is the cut-off, while $f_{n_1 n_2 n_3}$ stand for a fixed basis of functions. The standard procedure involves cosine function, however, it will be convenient to adopt a different choice. The solution to the eigenequation (87) is specified by fixing the unknown coefficients $c_{n_1,n_2,n_3}$. They can be determined by demanding the eigenequation to be satisfied at a lattice composed of fixed points. To illustrate this, let us restrict for simplicity to the one-dimensional case, rewriting the ansatz (88) as

$$f(x) = \sum_{n=1}^{N} c_n f_n(x). \tag{89}$$

The eigenequation reads

$$\hat{F}_{\alpha\gamma} f(x) = \lambda f(x). \tag{90}$$

Let $\{x_n\}_{n=1}^{N}$ stands for the lattice. For instance, one could consider Tchebychev's nodes. These are defined as roots of the Tchebychev polynomial of the first kind of the degree $n$ [24]. On the finite interval $[-1, 1]$, they read

$$x_n = \cos\left( \frac{2n-1}{2N} \pi \right), \quad n = 1, ..., N. \tag{91}$$

This can be extended on $[a, b]$ defining

$$x_n = \frac{a+b}{2} + \frac{b-a}{2} \cos\left( \frac{2n-1}{2N} \pi \right), \quad n = 1, ..., N. \tag{92}$$

Here, it is important that the number of points should match the number of coefficients $c_n$. At the lattice Equation (90) can be rewritten as

$$F_{nm}^{\alpha\gamma} c^m = \lambda f_{nm} c^m, \tag{93}$$

where $\vec{c} = (c^n) = \{c_1, ..., c_N\}$ is a vector built out of unknown coefficients, while $(f_{nm})$ and $(F_{nm})$ stand for $N \times N$ matrices defined as

$$f_{nm} := f_m(x_n), \quad F_{nm}^{\alpha\gamma} := \hat{F}_{\alpha\gamma} f_m(x)|_{x=x_n}. \tag{94}$$

Solving Equation (93) one guaranties the combination (89) satisfies the eigenequation (90) at $x = x_n$, $n = 1, ..., N$. Equation (93) is a generalized eigenequation: having specified matrices $(f_{nm})$ and $(F_{nm}^{\alpha\gamma})$ one obtains unknown coefficients $c_n$ and the eigenvalue $\lambda$ solving algebraic eigenequation (93). The coefficients specify approximate solution of the differential equation. The denser the grid $\{x_n\}$, the better the precision.

Now, we consider three-dimensional case. The first element of the construction is the functional basis $f_{n_1 n_2 n_3}(x_1, x_2, x_3)$. It turns out, a convenient choice is the basis

$$f_{n_1 n_2 n_3}(x_1, x_2, x_3) = \sin\left(1 + \frac{\ln|x_1|}{n_1} + \frac{\ln|x_2|}{n_2} + \frac{\ln|x_3|}{n_3}\right). \tag{95}$$

We can now search for solutions to the eigenequation (87). Finding the numerical solution $f_{n_1 n_2 n_3}$ of Equation (87), one finds the solution to the eigenequation (69), given by Equation (84). In order to do so, consider a three-dimensional grid $\{x_n\} = \{x_1^{(n)}, x_2^{(n)}, x_3^{(n)}\}$. As we are interested in covering both positive and negative $x_i \in \mathbb{R}$ in (95), it is reasonable to allow negative $n_i$ (we exclude $n_i = 0$ because of the form of the right hand side of Equation (95)). The sum (89) becomes $\sum_{n=-N}^{-1} c_n f_n(x) + \sum_{n=1}^{N} c_n f_n(x)$. Due to the presence of logarithmic function in Equation (95), this choice should respect the fact that terms in Equation (95) become highly oscillating in the limit $x_i \to 0$. Hence, it is reasonable to make the grid denser close to zero. However, this is not the case for the original Tchebychev's nodes (92). This can be achieved adopting the following, modified Tchebychev's nodes:

$$x_n = b_\pm \left(1 + \cos\left(\frac{2n-1}{4N}\pi\right)\right), \quad n = N+1, ..., 2N, \tag{96}$$

where $b_\pm$ stands for two real parameters, positive $b_+$ and negative $b_-$. They provide respectively positive and negative nodes. Clearly, this holds for all three dimensions. Because of terms $|x_1 x_2 x_3|^\alpha \exp(-\gamma/2(|x_1| + |x_2| + |x_3|))$, the function (84) vanishes close to zero, $|x_I| \ll 1$, and close to infinity, $|x_I| \gg 1$. Therefore, one gets a good approximation restricting to a relatively small finite number of nodes. This justifies the choice (96). Having specified the grid and functional basis, we are ready to find the solutions. Choosing (It turns out that takeing $|b_-| \neq |b_+|$ significantly improves numerical precision.)

$$b_- = -3, \quad b_+ = 3.5, \quad N = 5, \tag{97}$$

and adopting the basis (95) with

$$\alpha = \frac{1}{2}, \quad \gamma = 1, \tag{98}$$

we get approximate solutions with discrete spectrum of positive and negative eigenvalues. Restricting to negative values and starting with the highest $\lambda$, one finds:

$$\begin{aligned}
\lambda \simeq &- 2.193, -2.193, -6.470, -6.470, -14.34, -17.71, -27.34, -27.34, \\
&- 32.05, -36.26, -39.65, -47.50, -47.50, -62.62, -62.62, -63.83, \\
&- 74.97, -102.0, -110.7, -125.1, -125.1, -125.7, -155.9, -249.6, \\
&- 249.6, -325.3, -325.3, -358.0, -358.0
\end{aligned} \tag{99}$$

For instance, choosing the third eigenvalue one finds the solution $\psi_s$, and the corresponding numerical error given by Equation (81):

$$\psi_s: \quad \lambda_s = -6.470, \quad E[\psi_s] \simeq 2.86 \times 10^{-6}. \tag{100}$$

Here $\psi_s$ stands for a normalized function found by renormalization of the numerical solution $\psi_s \to \psi_s / \|\psi_s\|^{1/2}$. We have obtained a fairly good precision despite considering a small grid. More precisely, taking $N = 5$ means we adopted ten points per dimension (the whole three-dimensional lattice is composed of 1000 points). The rationale for this is the following. First, the functions (95) provides a good basis in the sense that a combination involving a small number of terms results in a good approximation to the solution of the eigenequation (in the sense the numerical error turns out to be small). This is because of the presence of logarithmic function; something that has already been observed discussing variational method. Second, the eigenfunction vanishes fast for $|x_I| \gg 1$, and so we can restrict our analysis to covering a small, finite region $x_I \in [b_-, b_+]$.

In addition to the symmetric function (95), one can consider the antisymmetric one, adopting the basis

$$f_{n_1 n_2 n_3}(x_1, x_2, x_3) = (\text{sign}(x_1) + \text{sign}(x_2) + \text{sign}(x_3)) \cdot$$
$$\cdot \sin\left(1 + \frac{\ln|x_1|}{n_1} + \frac{\ln|x_2|}{n_2} + \frac{\ln|x_3|}{n_3}\right). \tag{101}$$

Adopting the choices (97)–(98) leads to the following spectrum of negative eigenvalues:

$$\begin{aligned}
\lambda \simeq & -2.193, -2.193, -6.470, -6.470, -14.34, -17.71, -32.05, -36.26, \\
& -39.65, -47.50, -47.50, -63.83, -74.97, -96.98, -96.98, -102.0, \\
& -110.7, -125.7, -155.9, -249.6, -249.6, -325.3, -325.3, -358.0, -358.0.
\end{aligned} \tag{102}$$

Choosing, for instance, the first eigenvalue, one finds the antisymmetric solution $\psi_a$ and the corresponding error (81):

$$\psi_a: \quad \lambda_a = -2.193, \quad E[\psi_a] \simeq 2.68 \times 10^{-6}. \tag{103}$$

The functions $\psi_s$ and $\psi_s$ are orthogonal and they both were constructed as normalized.

### 3.3. Imposition of the Dynamical Constraint

Equation (65) is the Schrödinger-like equation corresponding to the classical dynamics defined by Equation (13)–(14). However, the latter is constrained by the condition $H = 0$, with $H$ given by (7). The Dirac quantization scheme applied in this paper consists in mapping the classical constraint to the quantum constraint $\hat{H} = 0$, which according to Equation (70) reads:

$$\hat{H}\Psi(\phi, x) := \left[E\left(\frac{\partial}{\partial \phi}\right) + \hat{W}\right]\Psi(\phi, x) = 0. \tag{104}$$

Therefore, not all solutions to (65) are physical but only the ones satisfying (104). It turns out, however, that the solution to (104), in the form (66) with $\omega_\lambda$ defined by (74)–(75), can only be the trivial one $\Psi(x) = 0$. To address this difficulty, we propose to impose, instead of (104), the weak form of the Dirac condition:

$$\langle \Psi | \hat{H}\Psi \rangle =: \langle \hat{H} \rangle_\Psi = 0, \tag{105}$$

which has to be satisfied by a given linear combination of the products of eigenfunctions

$$\Psi(\phi, x) = \fint_\lambda \omega_\lambda(A_\lambda, B_\lambda; \phi)\, \psi_\lambda(x), \tag{106}$$

where $\omega_\lambda$ are defined by (74)–(75) (up to arbitrary constants $A_\lambda$ and $B_\lambda$), and $\psi_\lambda$ is determined numerically via (76) and (84). The symbol $\fint_\lambda$ denotes summation or integration depending on the solutions to the eigenequations (68)–(69).

Since $\hat{W}$ is a Hermitian operator, we have $\langle \psi_{\lambda'} | \hat{W} | \psi_\lambda \rangle = \lambda \, \delta(\lambda', \lambda)$, with $\lambda \in \mathbb{R}$, so Equation (105) takes the form

$$\langle \hat{H} \rangle_\Psi = \sum_\lambda\!\!\!\!\!\!\!\!\int \omega_\lambda^*(A_\lambda, B_\lambda; \phi) \left( \lambda - \frac{\kappa}{2} \frac{d^2}{d\phi^2} \right) \omega_\lambda(A_\lambda, B_\lambda; \phi) = 0 \,. \tag{107}$$

For the case $\kappa\lambda \neq 1/2$, Equation (107) leads to

$$\sum_\lambda\!\!\!\!\!\!\!\!\int \omega_\lambda^*(A_\lambda, B_\lambda; \phi) \big( \omega_\lambda(A_\lambda, B_\lambda; \phi) + i\sqrt{2\kappa\lambda - 1} \, \omega_\lambda(A_\lambda, -B_\lambda; \phi) \big) = 0 \,, \tag{108}$$

whereas for the case $\kappa\lambda = 1/2$ we get

$$\omega_\lambda^*(A_\lambda, B_\lambda; \phi) \, \omega_\lambda(\kappa A_\lambda, \kappa B_\lambda + i A_\lambda; \phi) = 0 \,. \tag{109}$$

In what follows we consider $\kappa = 1$ and $\lambda < 1/2$, in which case $2\kappa\lambda - 1 < 0$ so that Equation (74) presents an oscillatory solution.

For $\lambda < 0$, one has

$$1 + \sqrt{|2\lambda - 1|} > 0 \quad \text{and} \quad 1 - \sqrt{|2\lambda - 1|} < 0 \,. \tag{110}$$

Equation (108) leads to the condition

$$\sum_\lambda\!\!\!\!\!\!\!\!\int [(1 - \sqrt{|2\lambda - 1|})|A_\lambda|^2 + (1 + \sqrt{|2\lambda - 1|})|B_\lambda|^2] = 0 \,, \tag{111}$$

where $A_\lambda B_\lambda = 0$. Assuming the orthonormality condition $\langle \psi_{\lambda'} | \psi_\lambda \rangle = \delta(\lambda', \lambda)$, we get

$$\sum_\lambda\!\!\!\!\!\!\!\!\int |\omega_\lambda(A_\lambda, B_\lambda; \phi)|^2 = 1 \,. \tag{112}$$

Equations (110)–(112) lead to the condition

$$\sum_{\lambda \in \mathcal{O}_1}\!\!\!\!\!\!\!\!\!\!\int |A_\lambda|^2 + \sum_{\lambda \in \mathcal{O}_2}\!\!\!\!\!\!\!\!\!\!\int |B_\lambda|^2 = 1 \,, \quad \text{where} \quad \mathcal{O}_1 \cap \mathcal{O}_2 = \varnothing \,. \tag{113}$$

Let us consider a special solution including only two eigenvalues $\lambda \neq \lambda'$. In such a case Equations (111)–(113) give

$$\left( 1 - \sqrt{|2\lambda - 1|} \right) |A_\lambda|^2 + \left( 1 + \sqrt{|2\lambda' - 1|} \right) |B_{\lambda'}|^2 = 0 \,, \tag{114}$$

and

$$|A_\lambda|^2 + |B_{\lambda'}|^2 = 1 \,. \tag{115}$$

The solution to (114)–(115) reads

$$|A_\lambda|^2 = \frac{1 + \sqrt{|2\lambda' - 1|}}{\sqrt{|2\lambda - 1|} + \sqrt{|2\lambda' - 1|}} \,, \quad |B_{\lambda'}|^2 = \frac{\sqrt{|2\lambda - 1|} - 1}{\sqrt{|2\lambda - 1|} + \sqrt{|2\lambda' - 1|}} \,. \tag{116}$$

Therefore, one of the possible solutions to the constraint (105) has the form

$$\Psi(\phi, x) = \omega_\lambda(A_\lambda, B_\lambda; \phi) \, \psi_\lambda(x) + \omega_{\lambda'}(A_{\lambda'}, B_{\lambda'}; \phi) \, \psi_{\lambda'}(x) \,, \tag{117}$$

which is defined by the specification of any pair of $\lambda \neq \lambda'$.

## 4. Quantum Spikes

The expectation values of our basic observables (42) in a state described by the wave-function (106) (satisfying the weak Dirac condition (105)) read

$$
\begin{aligned}
\langle \hat{C}_I \rangle (\phi) &= \int_{\mathbb{R}^3_+} d\nu(x_1, x_2, x_3) \Psi^\star(\phi, x_1, x_2, x_3) \hat{C}_I \Psi(\phi, x_1, x_2, x_3) \\
&= \sum_{\lambda_1, \lambda_2}\!\!\!\!\!\!\!\!\!\! \oint \; \omega^\star_{\lambda_1}(A_{\lambda_1}, B_{\lambda_1}; \phi)\, \omega_{\lambda_2}(A_{\lambda_2}, B_{\lambda_2}; \phi) \, \langle \psi_{\lambda_1} | \hat{C}_I | \psi_{\lambda_2} \rangle ,
\end{aligned}
\tag{118}
$$

and

$$
\langle \hat{D}_I \rangle (\phi) = \sum_{\lambda_1, \lambda_2}\!\!\!\!\!\!\!\!\!\! \oint \; \omega^\star_{\lambda_1}(A_{\lambda_1}, B_{\lambda_1}; \phi)\, \omega_{\lambda_2}(A_{\lambda_2}, B_{\lambda_2}; \phi) \, \langle \psi_{\lambda_1} | \hat{D}_I | \psi_{\lambda_2} \rangle .
\tag{119}
$$

The coefficients $A_\lambda$, $B_\lambda$ occurring here can be fixed by imposing the initial conditions at a certain value of $\phi = \phi_0$ so that for the case $I = 1$ we have (Since the classical spike has been derived for the case $I = 1$, we stick to this case at the quantum level as well. However, the system is symmetric with respect to the choice of $I$ so that the same is true for two other cases.):

$$
\langle \hat{C}_1 \rangle (\phi_0) =: \tilde{x}_{C_1},
\tag{120}
$$

$$
\langle \hat{D}_1 \rangle (\phi_0) =: \tilde{x}_{D_1},
\tag{121}
$$

with $\tilde{x}_{C_1}, \tilde{x}_{D_1} \in \mathbb{R}$. In principle, $\phi_0$ can be arbitrary but we choose $\phi_0 = 0$ below, as we also did for classical variables in Figures 1–3.

In the next two subsections, we apply our numerical results for the wavefunction $\Psi$ to calculate (118) and (119).

### 4.1. Using the Results of the Variational Method

Let us first consider the solutions (79)–(80) and (82)–(83). We calculate

$$
(C_1) = \begin{pmatrix} 0 & c_0 + i\delta c \\ c_0 - i\delta c & 0 \end{pmatrix},
\tag{122}
$$

where $(C_1)_{ij} := \langle \psi_{\lambda_i} | \hat{C}_1 | \psi_{\lambda_j} \rangle$, and we get

$$
c_0 \simeq -0.0174, \quad \delta c \simeq -2.21 \cdot 10^{-4}.
\tag{123}
$$

The full wavefunction $\Psi(\phi, x)$ is given by Equation (117) with $\lambda = \lambda_1$ and $\lambda' = \lambda_2$. Due to the condition $A_\lambda B_\lambda = 0$ (cf. (111)), we may assume that e.g., $A_{\lambda_2} = B_{\lambda_1} = 0$. Then, $A_{\lambda_1}$ and $B_{\lambda_2}$ remain two independent complex parameters. Let us parameterize them as

$$
A_{\lambda_1} = |A_{\lambda_1}| e^{i\varphi_1}, \quad B_{\lambda_2} = |B_{\lambda_2}| e^{i\varphi_2}.
\tag{124}
$$

The absolute values $|A_{\lambda_1}|$, $|B_{\lambda_2}|$ are fixed by Equations (116):

$$
|A_{\lambda_1}| = \sqrt{\frac{1 + \sqrt{|2\lambda_2 - 1|}}{\sqrt{|2\lambda_1 - 1|} + \sqrt{|2\lambda_2 - 1|}}}, \quad |B_{\lambda_2}| = \sqrt{\frac{\sqrt{|2\lambda_1 - 1|} - 1}{\sqrt{|2\lambda_1 - 1|} + \sqrt{|2\lambda_2 - 1|}}}.
\tag{125}
$$

The considered numerical solutions (80) and (83) correspond, respectively, to the values $\lambda_1 \simeq -0.0821$ and $\lambda_2 \simeq -0.0957$. Since we are interested in the oscillatory case, we choose $\kappa = 1$, so that (74) gives us complex functions

$$
\omega_{\lambda_1}(\phi) = A_{\lambda_1} e^{i\left(\sqrt{|2\lambda_1 - 1|} - 1\right)\phi}, \quad \omega_{\lambda_2}(\phi) = B_{\lambda_2} e^{-i\left(\sqrt{|2\lambda_2 - 1|} + 1\right)\phi}.
\tag{126}
$$

The final form of the wavefunction reads

$$\Psi(\phi, x) = \omega_{\lambda_1}(\phi)\, \psi_{\lambda_1}(x) + \omega_{\lambda_2}(\phi)\, \psi_{\lambda_2}(x)\,. \tag{127}$$

Calculation of the expectation value (118) for the state (127) leads to the result

$$\langle \hat{C}_1 \rangle(\phi) = \beta \cos(\Delta\varphi + \chi\phi) + \delta\beta \sin(\Delta\varphi + \chi\phi)\,, \tag{128}$$

with the following parameters

$$\Delta\varphi := \varphi_1 - \varphi_2\,, \tag{129}$$

$$\beta := 2c_0 |A_{\lambda_1}||A_{\lambda_2}|\,, \quad \delta\beta = 2\delta c\, |A_{\lambda_1}||A_{\lambda_2}|\,, \tag{130}$$

$$\chi := \sqrt{|2\lambda_1 - 1|} + \sqrt{|2\lambda_2 - 1|}\,. \tag{131}$$

The values of $\beta$, $\delta\beta$ and $\chi$ for the case of (123) and $\lambda_1, \lambda_2$ mentioned below (125) are

$$\beta \simeq -0.0065\,, \quad \delta\beta \simeq -8.3 \cdot 10^{-6}, \quad \chi \simeq 2.17\,. \tag{132}$$

Meanwhile, the parameter $\Delta\varphi$ can be eliminated from Equation (128) by imposing on the latter the initial condition (120), which gives

$$\beta \cos(\Delta\varphi + \chi\phi_0) + \delta\beta \sin(\Delta\varphi + \chi\phi_0) = \tilde{x}\,, \tag{133}$$

where we simplified the notation by taking $\tilde{x} \equiv \tilde{x}_{C_1}$. Equation (133) allows to express $\Delta\varphi$ as a function of $\tilde{x}$. Assuming that $\phi_0 = 0$, we find two different solutions $\Delta\varphi^{(\pm)} = \Delta\varphi^{(\pm)}(\tilde{x})$:

$$\Delta\varphi^{(\pm)} = \mathrm{atan2}\left( \frac{\beta\tilde{x} \mp |\delta\beta|\sqrt{\beta^2 + \delta\beta^2 - \tilde{x}^2}}{\beta^2 + \delta\beta^2}, \frac{\delta\beta\tilde{x} \pm |\beta|\mathrm{sign}(\delta\beta)\sqrt{\beta^2 + \delta\beta^2 - \tilde{x}^2}}{\beta^2 + \delta\beta^2} \right)$$
$$+ 2n\pi\,, \quad n \in \mathbb{Z}\,, \tag{134}$$

where atan2(.,.) stands for two-argument arctangent function. Substituting (134) into Equation (128), we ultimately obtain

$$\langle \hat{C}_1 \rangle_{(\pm)}(\phi) = \cos(\chi\phi)\,\tilde{x} \mp \mathrm{sign}(\delta\beta) \sin(\chi\phi)\sqrt{\beta^2 + \delta\beta^2 - \tilde{x}^2}\,. \tag{135}$$

In order to detect quantum spikes, we now examine $\langle \hat{C}_1 \rangle_{(\pm)}$ for a fixed $\phi$ but as a function of $\tilde{x}$. The dependence on $\tilde{x}$ is trivial for $\phi = 0$. For $\phi \neq 0$, the domain is restricted to the interval $\tilde{x} \in [-\sqrt{\beta^2 + \delta\beta^2}, \sqrt{\beta^2 + \delta\beta^2}]$ and the function $\langle \hat{C}_1 \rangle_{(\pm)}(\tilde{x})$ has the non-trivial derivative:

$$\frac{d\langle \hat{C}_1 \rangle_{(\pm)}(\tilde{x})}{d\tilde{x}} = \cos(\chi\phi) \pm \frac{\mathrm{sign}(\delta\beta) \sin(\chi\phi)\,\tilde{x}}{\sqrt{\beta^2 + \delta\beta^2 - \tilde{x}^2}}\,. \tag{136}$$

In particular, at $\tilde{x} = 0$ we have

$$\left.\frac{d\langle \hat{C}_1 \rangle_{(\pm)}(\tilde{x})}{d\tilde{x}}\right|_{\tilde{x}=0} = \cos(\chi\phi)\,, \tag{137}$$

which vanishes if

$$\chi\phi = \frac{\pi}{2} + k\pi\,, \quad k \in \mathbb{Z}\,. \tag{138}$$

The second derivative of Equation (135) reads

$$\frac{d^2 \langle \hat{C}_1 \rangle_{(\pm)}(\tilde{x})}{d\tilde{x}^2} = \pm \frac{(\beta^2 + \delta\beta^2)}{(\beta^2 + \delta\beta^2 - \tilde{x}^2)^{3/2}} \mathrm{sign}(\delta\beta) \sin(\chi\phi)\,. \tag{139}$$

Equations (137)–(139) show that, depending on the values of $\chi\phi$ and $\delta\beta$, $\langle\hat{C}_1\rangle_{(\pm)}$ reach a local maximum or minimum at $\tilde{x} = 0$. In particular, taking (132) and $k = 0$ in (138), one finds that this happens for $\phi \simeq 0.72$. We consider such a phenomenon to be the quantum analogue of a classical spike. In general, these quantum spikes occur only at specific moments of time $\phi$, belonging to a periodic discrete set with the period $\Delta\phi = \pi/\chi \simeq 1.45$, determined by Equation (138).

The function $\langle\hat{C}_1\rangle_{(\pm)}(\tilde{x})$ is shown in Figure 4. In both cases spikes occurring for $\phi \simeq 0.72$ are represented by solid lines.

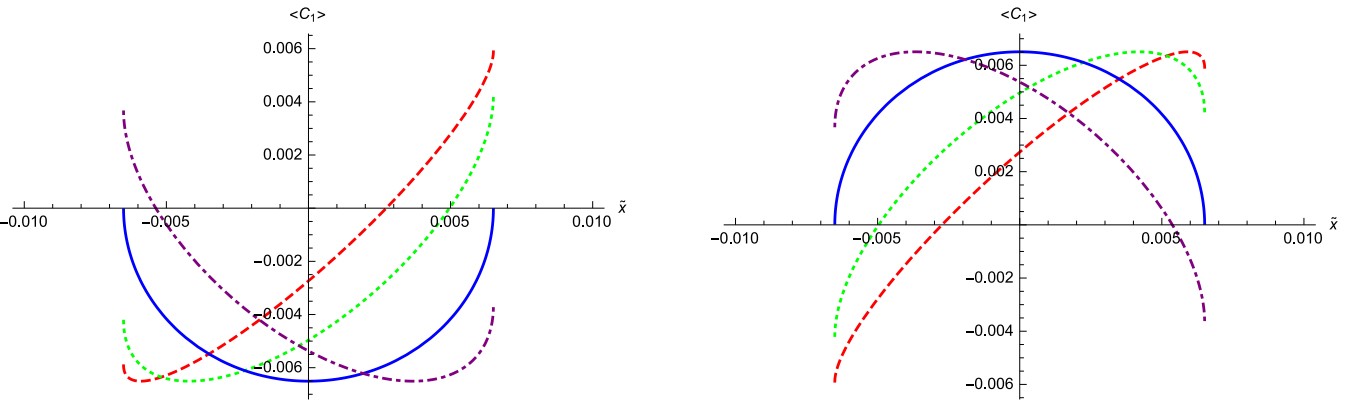

**Figure 4.** $\langle\hat{C}_1\rangle_{(-)}$ (left) and $\langle\hat{C}_1\rangle_{(+)}$ (right) for evolution parameters $\phi = 0.2$ (dashed), $\phi = 0.4$ (dotted), $\phi = 0.72$ (solid) and $\phi = 1$ (dash-dotted).

Having calculated $\langle\hat{C}_1\rangle$, we can repeat the above analysis for $\langle\hat{D}_1\rangle$. The matrix of elements $(D_1)_{ij} := \langle\psi_{\lambda_i}|\hat{D}_1|\psi_{\lambda_j}\rangle$ has the form

$$(D_1) = \begin{pmatrix} d_1 & 0 \\ 0 & d_2 \end{pmatrix} \tag{140}$$

where

$$d_1 \simeq -5.71 \cdot 10^{-6}, \quad d_2 \simeq 7.35 \cdot 10^{-5}. \tag{141}$$

Analogously to (128), we obtain

$$\langle\hat{D}_1\rangle = \frac{d_1 - d_2 + d_1\sqrt{|1 - 2\lambda_2|} + d_2\sqrt{|1 - 2\lambda_1|}}{\sqrt{|1 - 2\lambda_1|} + \sqrt{|1 - 2\lambda_2|}} \tag{142}$$
$$\simeq -2.83 \cdot 10^{-6}.$$

Here, $\langle\hat{D}_1\rangle = $ const because there are no off-diagonal components in the matrix (140). More precisely, the off-diagonal components are non-zero, but are small of order $10^{-20}$. This result is almost unaffected by change of the order $N$ of the numerical approximation and off-diagonal components are actually becoming smaller with growing (For instance, for $N = 8$ one finds them to be equal $\simeq 7.7 \cdot 10^{-20}$, while for $N = 10$ one gets $\simeq 2.2 \cdot 10^{-20}$.) $N$. In conclusion, the numerical results indicate that $\langle\hat{D}_1\rangle$ does not evolve with time $\phi$.

It is also worth stressing that, at least in the case of restriction to a superposition of two eigenstates (117), the presence of a spike-like structure, associated with the observable $\hat{C}_1$ is unaffected by the choice of a pair of numerical solutions, i.e., one symmetric and one antisymmetric wavefunction. Adopting different ones modifies the values of coefficients $\beta$, $\delta\beta$, $\chi$ but one can still find the value of $\phi$ corresponding to the quantum spike. In the case of $\hat{C}_1$ the latter is given by Equation (138); this is a simple function of two eigenvalues $\lambda_1$, $\lambda_2$. In fact, the crucial requirement for the occurrence of spikes in the presence of non-zero components in the matrix $(C_1)$.

### 4.2. Using the Results of the Spectral Method

One can now perform the analogous analysis for numerical solutions obtained via the spectral method. Taking $\psi_1 = \psi_s$, $\lambda_1 = \lambda_s$ given by Equation (100) and $\psi_2 = \psi_a$, $\lambda_2 = \lambda_a$ given by Equation (103), we obtain

$$(C_1) = \begin{pmatrix} 0 & c_0 \\ c_0 & 0 \end{pmatrix}, \tag{143}$$

where $c_0 = -1.28 \cdot 10^{-5}$, while $(D_1)_{ij} = 0$, $\forall i, j$. The matrices $(C_1)$ and $(D_1)$ are defined as before in Equations (122) and (140) but in contrast to the former case, the numerical solutions do not contain imaginary terms. For this reason, we have $(D_1) = 0$, as well as $\delta c = 0$. On the other hand, we can achieve the much better precision.

Following the same steps as described in the previous subsection, one can again express the expectation value $\langle \hat{C}_1 \rangle$ as a simple function of $\phi$ (cf. (128)):

$$\langle \hat{C}_1 \rangle(\phi) = \beta \cos(\Delta\varphi + \chi\phi), \tag{144}$$

where $\Delta\varphi$, $\beta$ and $\chi$ are given by Equations (129)–(131). The numerical values of the latter two constants for the considered case of (100) and (103) are

$$\beta \simeq 1.28 \cdot 10^{-5}, \quad \chi \simeq 6.05. \tag{145}$$

Similarly to the Equation (133), we eliminate the parameter $\Delta\varphi$ by imposing the boundary condition (120), which now becomes

$$\beta \cos(\Delta\varphi + \chi\phi) = \tilde{x}. \tag{146}$$

Solving this equation for $\Delta\varphi$, one finds two solutions (as in (134) before)

$$\Delta\varphi^{(\pm)} = \pm \arccos\left(\frac{\tilde{x}}{\beta}\right) + 2n\pi, \quad n \in \mathbb{Z}. \tag{147}$$

Substitution of $\Delta\varphi = \Delta\varphi^{(\pm)}$ into Equation (144) finally gives

$$\langle \hat{C}_1 \rangle_{\pm}(\phi) = \tilde{x} \cos(\chi\phi) \mp \beta \sqrt{1 - \frac{\tilde{x}^2}{\beta^2}} \sin(\chi\phi). \tag{148}$$

The two obtained solutions are depicted in Figure 5. Quantum spikes (represented by solid lines) occur in both cases at time $\phi \simeq 0.26$ and are periodic in $\phi$, with the period $\Delta\phi = \pi/\chi \simeq 0.52$.

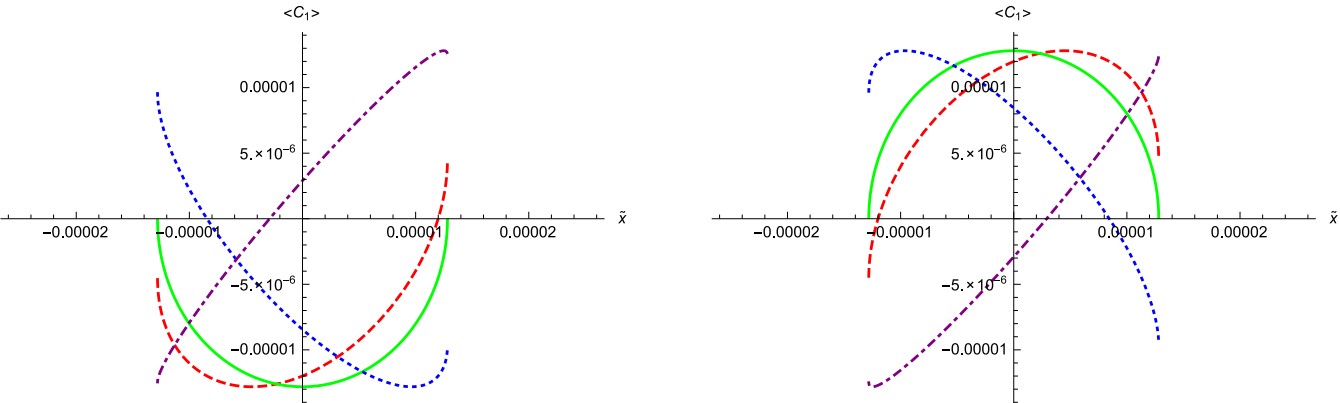

**Figure 5.** $\langle \hat{C}_1 \rangle_{(-)}$ (left) and $\langle \hat{C}_1 \rangle_{(+)}$ (right) for evolution parameters $\phi = 0.2$ (dashed), $\phi = 0.26$ (solid), $\phi = 0.4$ (dotted), $\phi = 1$ (dash-dotted).

The existence of spikes is ensured by the presence of non-zero elements in the matrix $(C_1)$. Starting with a different pair of symmetric and antisymmetric functions $\psi_s$ and $\psi_a$, corresponding to different eigenvalues $\lambda_1$ and $\lambda_2$, one also expects to find spikes (unless $(C_1)_{ij} = 0$, $\forall i, j$).

## 5. Conclusions

In this paper, we have attempted to uncover the existence of quantum strange spikes (in short, quantum spikes), i.e., certain distinctive features in the quantum evolution of the considered gravitational system. Such features are expected to be analogs of steep structures, called by us strange spikes, that arise in the corresponding classical evolution. Figure 1 shows that the classical spikes presented in Figure 2 (and previously in the paper [16]) are not apparent effects of the projection of a 3D plot into a 2D plot, but real structures. Furthermore, it seems that an extremum (maximum, minimum) or an inflection point, occurring locally at the classical level, may turn into a similar structure at the quantum level. Quantization does not need to suppress the classical spikes as it was preliminarily concluded in [16].

Let us discuss the latter claim in more detail. On the basis of numerical results, we conjecture that a quantum spike is an extremum in the evolution of the expectation value of a given quantum observable. The inflection-type classical spike $C_1(\tilde{x}, \phi)$ presented in Figures 1 and 2 becomes the extremum-type quantum spike presented in Figures 4 or 5. Another difference is that our classical spike is a monotonic function of time, whereas the corresponding candidate for a quantum spike is periodic. To be specific, we have restricted our analyses to just one quantum observable $\hat{C}_1$ (the results for $\hat{D}_1$ have insufficient accuracy).

Compared to the classical spikes, quantum spikes differ in two ways. Namely, they are rather mild and periodic in time. At the quantum level, classical structures become smoother and of a slightly different type, being specific only to discrete moments in time. Nevertheless, it can be conjectured that classical spikes survive quantization in this sense.

Constructing solutions of the quantum evolution, we have applied the variational and spectral methods, which are quite different. These methods are not only different conceptually but also use different bases of auxiliary functions. Still, the obtained quantum spikes are quite similar. In particular, they are periodic in time. The two completely dissimilar methods lead to similar structures, which may serve as the robustness test of our results.

For simplicity, we have identified quantum spikes by making use of only two classes (obtained by two different numerical methods) of solutions to the quantum dynamics. Many other classes of solutions are possible and hence other types of quantum spikes could exist. Some of them might look similar to classical spikes and be monotonic functions of time. However, increasing the number of solutions in a wave packet makes the construction

much more technically involved. This is the reason we have restricted ourselves to a pair of numerical solutions, but it was sufficient to test the method.

Another issue that we need to stress is our application of a classical massless scalar field in the role of a clock at both the classical and quantum levels. The scalar field is not coupled to the gravitational degrees of freedom in the Hamiltonian (7) and hence such treatment is justified. In fact, this allows us to avoid the inconsistency that is commonly ignored in literature: the situation when time is a parameter in the classical theory but a quantized variable in the quantum theory. In both cases, it should be just the same evolution parameter. In this paper, we did not wish to consider the quantization of time.

The implementation of the dynamical constraint at the quantum level has been performed in the weak sense. Such a way of imposing constraints is practiced in other branches of quantum physics and quantum chemistry, especially in variational methods (see, e.g., [25–27] and references therein).

A study of the corresponding issues in the case of inhomogeneous spacetimes would be highly interesting since they naturally favor structure formation. Different to our strange spikes, the spikes found in such spacetimes (see [7–15] and references therein) have never been quantized. Let us also stress that we do not investigate the possible relation between the latter "inhomogeneous" spikes and our "homogeneous" ones, as it is beyond the scope of the present paper.

When we think about the real spikes, which might occur in the early observed universe, we rather think in terms of possible structures in spacetime. In contrast, the spikes that we study in this paper arise in the phase space of the Hamiltonian framework. The path from the dynamics in phase space to the dynamics in spacetime is complicated due to the Hamiltonian constraint. Apart from this, the truncation of the full system to the homogeneous sector considered in [6], which underlies our paper, introduces additional complexity. Thus, the interpretation of our spikes in terms of the spikes in spacetime is rather difficult. These difficulties are enhanced by the procedure of quantization. We postpone the examination of quantum spikes that are described directly in spacetime to our future work on the quantization of spikes known in the context of Gowdy space [7].

Our paper is about the possible existence of quantum spikes. The theoretical framework has been established but much more effort is necessary to prove that quantum spikes are a generic feature of the quantum gravitational systems. In particular, the preliminary results presented here could be extended by the further examination of the eigenequation problem (69). New classes of its solutions could lead to new types of quantum spikes. This extension of our research definitely requires making use of sophisticated analytical and numerical tools so that is far from being complete. More activity in this direction is needed.

**Author Contributions:** Authors contributed equally to this manuscript. All authors have read and agreed to the published version of the manuscript.

**Funding:** This research received no external funding.

**Acknowledgments:** We would like to thank Vladimir Belinski, Piotr Garbaczewski, David Garfinkle, Woei Chet Lim, David Sloan, and Claes Uggla for helpful discussions.

**Conflicts of Interest:** The authors declare no conflict of interest.

## Appendix A. Numerical Solutions

All coefficients have been found using Mathematica computer software.

The coefficients $c_{n_1 n_2 n_3}$ of $\psi_S$ corresponding to (79) read:

$$
\begin{aligned}
&c_{000} = 0.0000971398, c_{001} = -0.0000668377, c_{002} = 0.0000576278, \\
&c_{011} = 0.000145685, c_{003} = -0.0000408203, c_{012} = -0.000175153, \\
&c_{111} = -0.000093447, c_{004} = 5.14783 \times 10^{-6}, c_{013} = 0.000120379, \\
&c_{022} = 3.38894 \times 10^{-6}, c_{112} = 0.000204403, c_{005} = 6.14349 \times 10^{-6}, \\
&c_{014} = -0.0000357561, c_{023} = -0.0000709548, c_{113} = -0.000189175, \\
&c_{122} = 0.0000118927, c_{006} = 0.0000130402, c_{015} = -7.68871 \times 10^{-6}, \\
&c_{024} = -0.000037775, c_{033} = -0.0000601953, c_{114} = 0.0000173017, \\
&c_{123} = -0.00011458, c_{222} = 0.0000716101, c_{007} = 9.86615 \times 10^{-6}, \\
&c_{016} = 0.0000427615, c_{025} = -0.0000538152, c_{034} = 0.0000404286, \\
&c_{115} = 0.000035388, c_{124} = 0.000202186, c_{133} = 0.0000730395, \\
&c_{223} = 0.00011985, c_{008} = 3.4044 \times 10^{-6}, c_{017} = 0.0000249608, \\
&c_{026} = -0.0000690364, c_{035} = -0.0000391754, c_{044} = -0.0000453004, \\
&c_{116} = -0.000104671, c_{125} = -0.0000730649, c_{134} = -0.0000535717, \\
&c_{224} = 0.0000549831, c_{233} = -0.0000756014, c_{009} = 5.96683 \times 10^{-7}, \\
&c_{018} = 5.7956 \times 10^{-6}, c_{027} = -0.0000289886, c_{036} = -0.0000465147, \\
&c_{045} = -0.0000499186, c_{117} = -0.0000613235, c_{126} = -0.000175011, \\
&c_{135} = -0.000150828, c_{144} = -0.0000280041, c_{225} = -0.000230914, \\
&c_{234} = 0.0000408805, c_{333} = 0.0000680184, c_{0010} = 4.36818 \times 10^{-8}, \\
&c_{019} = 5.0905 \times 10^{-7}, c_{028} = -4.19215 \times 10^{-6}, c_{037} = -0.0000128081, \\
&c_{046} = -0.0000245689, c_{055} = -0.0000213292, c_{118} = -9.56085 \times 10^{-6}, \\
&c_{127} = -0.0000502355, c_{136} = -0.000111775, c_{145} = -0.0000995701, \\
&c_{226} = -0.000193417, c_{235} = -0.000350751, c_{244} = -0.000167525.
\end{aligned}
\tag{A1}
$$

Similarly, the coefficients $c_{n_1 n_2 n_3}$ of $\psi_A$ corresponding to (83) read:

$$
\begin{aligned}
&c_{000} = 0.0000154524, c_{001} = -7.25196 \times 10^{-7}, c_{002} = -6.31044 \times 10^{-6}, \\
&c_{011} = -1.84189 \times 10^{-6}, c_{003} = 7.81442 \times 10^{-6}, c_{012} = 4.7519 \times 10^{-6}, \\
&c_{111} = -6.32555 \times 10^{-6}, c_{004} = -6.01093 \times 10^{-6}, c_{013} = -0.000016969, \\
&c_{022} = -6.35142 \times 10^{-6}, c_{112} = 8.55031 \times 10^{-6}, c_{005} = -7.06458 \times 10^{-7}, \\
&c_{014} = 0.000023153, c_{023} = -2.43363 \times 10^{-6}, c_{113} = -0.0000170932, \\
&c_{122} = -0.0000151945, c_{006} = -5.85877 \times 10^{-6}, c_{015} = -6.02818 \times 10^{-6}, \\
&c_{024} = 2.59351 \times 10^{-6}, c_{033} = 7.14048 \times 10^{-6}, c_{114} = 0.0000165033, \\
&c_{123} = 0.0000425176, c_{222} = 5.83009 \times 10^{-6}, c_{007} = -3.85805 \times 10^{-6}, \\
&c_{016} = -3.95188 \times 10^{-7}, c_{025} = -0.0000125786, c_{034} = -0.0000220554, \\
&c_{115} = -0.0000163389, c_{124} = -0.0000196522, c_{133} = -0.0000324696, \\
&c_{223} = -0.0000273951, c_{008} = -8.02081 \times 10^{-7}, c_{017} = 0.0000107185, \\
&c_{026} = 1.27489 \times 10^{-6}, c_{035} = 9.31717 \times 10^{-6}, c_{044} = 5.99047 \times 10^{-6}, \\
&c_{116} = -5.39372 \times 10^{-7}, c_{125} = -1.92239 \times 10^{-6}, c_{134} = 0.0000483597, \\
&c_{224} = 0.0000255206, c_{233} = 0.0000448492, c_{009} = -3.69269 \times 10^{-8}, \\
&c_{018} = 4.74694 \times 10^{-6}, c_{027} = 5.24472 \times 10^{-6}, c_{036} = 7.44385 \times 10^{-6}, \\
&c_{045} = 6.93398 \times 10^{-6}, c_{117} = 6.12731 \times 10^{-6}, c_{126} = 0.0000300494, \\
&c_{135} = -0.0000468598, c_{144} = -0.0000161307, c_{225} = -0.0000284781, \\
&c_{234} = -0.000107147, c_{333} = -0.0000231205, c_{0010} = 3.87598 \times 10^{-9}, \\
&c_{019} = 6.21427 \times 10^{-7}, c_{028} = 1.58561 \times 10^{-6}, c_{037} = 1.98172 \times 10^{-6}, \\
&c_{046} = 0.0000147037, c_{055} = -3.09361 \times 10^{-6}, c_{118} = 2.09284 \times 10^{-6}, \\
&c_{127} = 0.0000119165, c_{136} = 0.0000362559, c_{145} = 8.76622 \times 10^{-6}, \\
&c_{226} = 0.0000623312, c_{235} = 0.000147871, c_{244} = 0.000069584.
\end{aligned}
\tag{A2}
$$

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
