# Peer review of "Hunting for Gravitational Quantum Spikes"

_universe, doi:10.3390/universe7030049_

Round 1

Reviewer 1 Report

This paper is in principle sound and presents an interesting topic. It might be accepted after major revision following the lines below. In particular, the presentation and many technical details must be improved before reconsidering the manuscript. Also more information on the physical terms of the relevant aspects should be given.

List of problems:

1) in sec 2.1 I read the unbelievable "massless matter field". This is complete nonsense, context is not sets so this is the only possible interpretation of this phrase.

2) Overall, one may request from physicists to write in a style much more physical than mathematical. See, also 1) above. The Hamiltonian eq. (7) may be introduced to set some physical context, which term corresponds to kinetic energy, which to the potential, and possible interaction(s) etc.

3) Some problems with language, but especially those that make it hard to understand, e.g. line 88 "are the nonhyperbolic", articles should be set correctly here an there for better reading and understanding.

4) fig. 1 needs much bigger labels and axes. In my b/w plot I can hardly see anything even on a big screen. Captions must be improved to state what the reader should see in the various figures.

5) I guess the star should be a dagger in eq. (53).

Some additional points on the physics shown:

a) reading the paper I have been wondering whether there is any connection with the much studied freak waves in classical wave dynamics and quantum mechanics, see e.g. P. K. Shukla, I. Kourakis, B. Eliasson, M. Marklund, and L. Stenflo, Phys. Rev. Lett. 97, 094501 (2006); D. R. Solli, C. Ropers, P. Koonath, and B. Jalali, Nature 450 1054 (2007); M. Onorato, S. Residori, U. Bortolozzo, A. Montina, and F.T. Arecchi, Physics Reports 528 47 (2013). These excitations seem to form spontaneously in the dynamical evolution, similar to the spikes discussed here. Any comment would be interesting and boost the papers impact!

b) whether any structure will be smoothed by quantum mechanics, as compared to what by the way, classical mechanics?, depends a lot on the relative scales, in cosmology typically on the parameter hbar/mass, or just hbar in usual quantum mechanics. This regards comments on the paper in lines 136/7 and the conclusions. Somethings should be said here! More physics! See my point 1).

c) When speaking about numerical simulations, more information is required, in particular on the precise method and the precision and convergence obtained. Singular problems like expected spikes, are not easy to faithfully be reproduced. Please keep this in mind, when talking to an educated audience.

Reviewer 2 Report

The authors have revised the manuscript to address my comments upto some extent. Although I am still not fully convinced of the potential impact of the work presented, I think the paper is in now good shape for publication. 

Reviewer 3 Report

I found this original paper very stimulating, and its results potentially very relevant for the generation of structures in the Universe. It addresses a cutting-edge topic (the one on the generation of spikes in the classical and quantum theory), which although might have not been yet properly discussed in the literature, can be easily envisaged to receive an increasing attention in the community over the next future. The topic still requires further analyses and discussion in the community. Nonetheless, this first analysis is definitely worthy of publication in this journal, in the present form.

Round 2

Reviewer 1 Report

I thank the authors for considering my suggestions. The most serious of them have been implemented, in particular the major issue 1) that the paper is not adequate in its style for a physics community. 

This manuscript is a resubmission of an earlier submission. The following is a list of the peer review reports and author responses from that submission.

Round 1

Reviewer 1 Report

see my attached report

Reviewer 2 Report

The authors study the so called phenomena of spikes in the classical and a quantum dynamics of type A homogeneous Bianchi models. While the mathematical steps seem fine, the paper lacks a clear motivation. The physical significance of the spikes are not discussed at all, neither in the classical nor in the quantum theory. For these reasons and the ones mentioned below I cannot recommend the paper for publication. I would like to point out following two overall aspects which authors can consider while revising and resubmitting the paper somewhere else:

1. Classically, the spikes are present in the inhomogeneous spacetime when evolved in time, which further lead to BKL type of behaviour. The authors find a different kind of spike, which from figures 1 and 2 seem to be just non-monotonic spike behaviour in variables $D_I$ with respect to $x$. The evolution with respect to $\phi$ still seems monotonic and there is no spike there. So, I am a little confused with the meaning of spikes here and their physical significance. Why are they important and what kind of implications might they have for the evolution of spacetime itself? These questions will be answered by studying the behaviour of physical observables such as scale factor or Hubble rate with respect to time.

2. In the quantum theory, the authors show the spike in expectation value of $C_I$ with respect to the time $\phi$. This is quite different from the ones shown in Fig. 1 and 2. What happens to the classical spikes in the quantum theory? also, what is the significance of these spikes in the quantum theory? What do they mean in terms of the physical observables?

Moreover, the paper is badly written and sometimes hard to understand what is meant. There are strange Latex typos in lines 219, 228, 232 and many more which mix the math font with english.

I would like to reiterate again that I don’t have any issue with the mathematical steps. I just fail to see any significance of the results and any physical insight that can be drawn from the results.

Reviewer 3 Report

Even if I do not feel very qualified to judge this paper, I have the feeling that the presentation can be improved. For an outsider, the paper is almost not comprehensible. Hence, I suggest to introduce much better the question, the problem, and the methods to be used to attack this question. Also formally there are many problems with the paper, just to mention 2:

1) eq. (50) misses the identity (right!?)

2) from section 3.2.2, the text is often mixed with formula environment making it hardly readable.

I suggest a thorough reworking and improvement of the paper before we may decide about a possible publication.